



# The Langley Ratio method, a new approach for transferring photometer calibration from direct sun measurements

Antonio F. Almansa[1,2], África Barreto[2,4], Natalia Kouremeti[3], Ramiro González[4], Akriti Masoom[3], Carlos Toledano[4], Julian Gröbner[3], Rosa D. García[5,2], Yenny González[1,2], Stelios Kazadzis[3], Stéphane Victori[1], Óscar Álvarez[2], Virgilio Carreño[2], Victoria E. Cachorro[4], and Emilio Cuevas[2]

[1]Cimel Electronique, Paris, France
[2]Izaña Atmospheric Research Center (IARC), Agencia Estatal de Meteorología (AEMET), Santa Cruz de Tenerife, Spain
[3]Physikalisch-Meteorologisches Observatorium Davos, World Radiation Center (PMOD/WRC), Davos, Switzerland
[4]Atmospheric Optics Group of Valladolid University (GOA–UVa), Valladolid University, Valladolid, Spain
[5]TRAGSATEC, Madrid, Spain

**Correspondence:** Antonio F. Almansa (f-almansa@cimel.fr)

**Abstract.**

This article presents a new method for transferring calibration from a reference photometer, referred to as the "master", to a secondary photometer, referred to as the "field", using a synergetic approach when master and field instruments have different spectral bands. The method was first applied between a PFR, (Precision Filter Radiometer) instrument from the World Optical Depth Research and Calibration Center (WORCC) considered the reference by the WMO (World Meteorological Organization), and a CE318-TS photometer, the standard photometer used by AERONET (AErosol RObotic NETwork). These two photometers have different optics, sun-tracking systems and spectral bands. The Langley Ratio method (LR) proposed in this study was used to transfer calibration to the closest spectral bands for 1-minute synchronous data, for airmasses between 2 and 5, and was compared to the state of the art Langley calibration technique. The study was conducted at two different locations, Izaña Observatory (IZO) and Valladolid, where measurements were collected almost simultaneously over a six-month period under different aerosol regimes. In terms of calibration aspects, our results showed very low relative differences and standard deviations in the calibration constant transferred in Izaña from PFR to Cimel, up to 0.29 % and 0.46 %, respectively, once external factors such as different field-of-view between photometers or the presence of calibration issues were considered. However, these differences were higher in the comparison performed at Valladolid (1.04 %) and in the shorter wavelengths spectral bands (up to 0.78 % in Izaña and 1.61 % in Valladolid). Additionally, the LR method was successfully used to transfer calibrations between different versions of the CE318-T photometer, providing an accurate calibration transfer (0.17 % to 0.69 %) in the morning LRs, even when the instruments had differences in their central wavelengths ($\Delta\lambda$ up to 91 nm). Overall, our results indicate that the LR method is a useful tool not only for transferring calibrations but also for detecting and correcting possible instrumental issues. This is exemplified by the temperature dependence on the two Cimel UV spectral bands, which was estimated by means of the LR method to be $\sim$ -0.09x10$^{-2}$/° in the case of 380 nm and $\sim$ -0.03x10$^{-2}$/° in the case of 340 nm. This estimation served us to implement the first operative temperature correction on ultraviolet (UV) spectral bands.





# 1   Introduction

Solar photometry is widely considered an accurate technique for determining aerosol properties in the atmosphere, as demonstrated by several studies (Schmid and Wehrli, 1995; Holben et al., 1998; Wehrli, 2000; Takamura et al., 2004; Nakajima et al., 2020; Kazadzis et al., 2018a, b, among others). Photometer networks operate worldwide to monitor atmospheric aerosols, with the AErosol RObotic NEtwork (AERONET) (Holben et al., 1998; Giles et al., 2019), the Sky Radiometer Network (SKYNET) (Takamura et al., 2004; Campanelli et al., 2004), and the Global Atmosphere Watch-Precision Filter Radiometer (GAW-PFR) (Wehrli, 2000, 2005) the most important due to their extensive coverage and high standardization levels. These global networks commonly provide spectral aerosol optical depth (AOD) data, which is considered the most comprehensive measure for estimating the columnar aerosol load and is also a crucial variable in radiative-forcing studies (Kazadzis et al., 2018a). In fact, spectral AOD is considered an "essential climate variable" by various organizations, such as the Global Climate Observing System (GCOS) World Meteorological Organization (WMO) program (Bojinski et al., 2014), the GAW programme (WMO, 2003), or the European Space Agency Climate Change Initiative (Popp et al., 2016).

AERONET (https://aeronet.gsfc.nasa.gov) is a federated network that comprises various national and regional networks. It was established in 1998 with the Cimel CE318 radiometer as the reference instrument (Holben et al., 1998; Giles et al., 2019). AERONET was created with the aim of validating satellite products and improving the global characterization of atmospheric aerosols and water vapor (Holben et al., 1998). With over 600 stations, AERONET is considered the largest photometer network in the world and provides long-term series of aerosol properties (Nyeki et al., 2012; Cuevas et al., 2019; Nakajima et al., 2020; Karanikolas et al., 2022). It also provides near-real-time (NRT) products that are useful for satellite validation (Omar et al., 2013; Sayer et al., 2017, 2019; Sogacheva et al., 2020; Chen et al., 2022, among others), as well as routine and retrospective climate model validation and model assimilation (Rubin et al., 2017; Randles et al., 2017; Benedetti et al., 2018; Gueymard and Yang, 2020; Mortier et al., 2020). AERONET reference photometers are calibrated at two high-mountain sites: Izaña Observatory (Tenerife, Spain) and Mauna Loa Observatory (Hawaii, USA).

GAW-PFR is a network of PFR radiometers (Wehrli, 2000, 2005, 2008a, b) that has been designated as the primary WMO Reference Centre for AOD measurements through the World Optical Depth Research and Calibration Centre (WORCC) at the Physikalisch-Meteorologisches Observatorium Davos, World Radiation Center (PMOD/WRC) (WMO, 2005; Kazadzis et al., 2018b). Currently, more than 40 stations around the world operate within this network, providing long-term aerosol observations, with 12 of them designated as core GAW-PFR sites by the WMO Scientific Advisory Group for aerosols. There are other associated stations currently providing PFR data which are not part of GAW, as is the case of Valladolid data used in the present study. There is overlapping between some GAW and AERONET/SKYNET stations, which allows PMOD/WRC not only to provide reference observations with their uncertainty estimation to contribute to GAW and GCOS but also to develop a strategy to ensure the traceability of AOD between different networks and merge their aerosol observations into a global dataset. GAW-PFR calibration chain, described in Kazadzis et al. (2018b), consists of a triad of instruments that continuously measure at PMOD/WRC, in addition to three portable transfer radiometers to guarantee traceability of AERONET with the world reference of the WORCC. These three radiometers operate in four sites: Mauna Loa, Izaña Observatory (IZO), Valladolid and



OHP (Observatoire de Haute-Provence, France), the last three lately as part of the ACTRIS (Aerosol Clouds and Trace Gases Research Infraestructure) calibration of aerosol remote sensing collaboration on ACTRIS-WMO sun-photometer calibration links.

Although solar photometry is a mature technique for retrieving aerosol products from solar direct measurements, calibrat-
ing sun-photometers still requires significant scientific effort in current photometric networks. Instrumental issues, such as decaying sensitivity of detectors and filters, or temperature-dependent problems of detectors and filters, also play a crucial role in accurately monitoring aerosol optical depth (AOD). However, as stated by Forgan (1994); Kazadzis et al. (2018a), the most significant challenge in solar photometry is calibration, namely the inability to relate sun-photometer observations accurately to irradiance standards (in the case of an absolute calibration method), or the uncertainty involved in estimating the exo-atmospheric signal or calibration constant ($V_{0,\lambda}$) through the Langley extrapolation technique. Recent efforts to address this issue have been presented by Kouremeti et al. (2022), focusing on linking AOD calibration standards to SI traceable ones. AOD calculation is highly sensitive to these factors, especially to calibration errors, since it is a small quantity that cannot be measured directly. As highlighted by Cachorro et al. (2008); Kazadzis et al. (2014), among others, an error of 1% in the calibration constant $V_{0,\lambda}$ leads to an error of 0.01 in AOD for an air mass equal to 1.

In this paper, we present a new methodology specifically designed to be applied when the calibration transference is carried out between two photometers with different spectral bands in terms of central wavelength ($\lambda_c$) or Full-Width-at-Half-Maximum (FWHM). The so-called Langley-Ratio method (LR) has been conceived as a robust calibration method, which is a mixture of the two most commonly used calibration techniques: Langley and Ratio calibration techniques. The Langley calibration method involves performing photometric measurements under very stable atmospheric conditions to estimate the zero-air-mass voltage (or calibration constant, $V_{0,\lambda}$) by extrapolation using the Bouguer-Lambert-Beer Law. This is an accurate method, with calibration uncertainty expected to be $\sim 0.25$-$0.5\%$, as stated in Toledano et al. (2018), which requires long observational periods (typically one or two months) usually performed at high-altitude stations. The Ratio calibration is a cross-calibration transference technique comparing at-ground voltage ratios between field and reference (master) instruments, normally performed around noon. This method assumes that the total optical depth of coincident measurements are equal, which only happens if the calibration is performed with instruments with quasi-coincident spectral bands. The Ratio cross-calibration is a faster calibration method (it usually takes some weeks) but uncertainty in this case is higher (less than 1% as stated in Holben et al. (1998); Giles et al. (2019)). LR can be considered as an extension of the Ratio calibration technique by considering the effect of different spectral bands in terms of $\lambda_c$ or FWHM of the two photometers. Therefore, LR is able to accomplish the calibration transference between different instruments (such as PFR and Cimel) but also between similar instruments with slight differences in some spectral bands. This is the case of the Cimel CE318-TS photometers, where variations in $\lambda_c$ and FWHM between instruments can have a critical impact in the UV range, or in the calibration of the CE318-TV12-OC (Zibordi et al., 2021). This issue was partially addressed by Fargion et al. (2001), who proposed a modified cross-calibration methodology within AERONET specifically for spectral bands centered at different wavelengths.





Coincident GAW-PFR and AERONET-Cimel photometric observations in Izaña GAW core station and in Valladolid (PFR associated station) have been used in this paper to analyze the performance of this new LR calibration method as well as its main advantages and disadvantages.

## 2   Instrumental description

### 2.1   Cimel Photometer CE318-T

The Cimel CE318-TS is the standard instrument in AERONET, a radiometer manufactured by Cimel Electronique that measures direct solar and lunar radiation together with sky radiance using a sensing head mounted on a two-axis tracker and a control unit (Holben et al., 1998; Barreto et al., 2016). The sensing head is equipped with two types of detectors, Silicon photodiode and InGaAs (Indium Gallium Arsenide) detector. The detectors are filtered by a set of optical bandpass filters with different wavelengths centred at 340, 380, 440, 500, 675, 870, 940, 1020 and 1640 nm, with an FWHM of 2 nm for 340 nm, 4 nm for 380 nm, 25 nm for 1640, and 10 nm for the rest of the channels. The 1020 nm spectral band is measured twice, using two detectors, hereafter referred to as the 1020 and 1020i measurements. The tolerance interval for central wavelength and FWHM is $\pm$ 0.5 nm, which means that the spectral response might vary between Cimel photometers, even in the case of those with the same version and model. The field of view (FOV) of the instrument is approximately 1.3° (Torres et al., 2013), and a removable collimator is used to minimize stray light when sky radiance is measured.

Solar tracking is performed using time-based ephemerides, latitude, and longitude and is enhanced using a four-quadrant sensor present in the sensor head. According to Torres et al. (2013), the pointing error of the Cimel CE318 is less than 0.1°. The measurements of direct solar radiation for all the current filters are performed in triplets, consisting of sequences of three measurements taken every 30 seconds, with a total duration of one minute for each triplet. Direct solar/lunar measurements are performed every 15 minutes between 9 a.m. and 3 p.m. local time, with a variable frequency depending on the air mass outside those hours. During periods when the radiometer is not in use, the tracker is positioned facing downwards, to minimize any potential damage to the optical filters from solar radiation. Microphysical and optical parameters of the aerosols can be retrieved with spectral radiance measurements together with the AOD, AERONET uses the code developed by Dubovik and King (2000), with some improvements introduced in Dubovik et al. (2002, 2006).

Typical uncertainties in AOD products for reference instruments are expected to range between 0.002 and 0.009, which are higher for shorter wavelengths. Uncertainty in AOD values in the case of field instruments are expected to be higher, between 0.010 and 0.020, with higher values for ultraviolet (UV) spectral bands (Holben et al., 1998; Eck et al., 1999).

The CE318-TV12-OC is another version of the CE318-T photometer specifically designed for ocean color applications. Equipped with 12 optical filters centred at 400, 412.5, 442.5, 490, 510, 560, 620, 665, 779, 865, 937, and 1020 nm, this photometer is utilized by AERONET Ocean Color (AERONET-OC) global network, installed on offshore platforms to measure the radiance emerging from the sea, a critical parameter for satellite ocean colour validation (Zibordi et al., 2021).





Cimel data used in this study has been screened for clouds by means of matching our dataset with AERONET version 3
level 2.0 quality-assured data. This technique ensures that our data are free of clouds, are filtered by instrumental problems and
the final post-field instrument calibration has been applied.

## 2.2  PFR Photometer

The PFR (Wehrli, 2000, 2005, 2008a, b) is an instrument manufactured by PMOD/WRC consisting of a sensor head and a
control unit. Unlike the Cimel CE318, it does not have its own tracker, and is usually mounted on commercial trackers that
always point towards the sun. The sensor head has four independent channels with bandpass optical filters centred at 368, 412,
500, and 862 nm, with a 5 nm FWHM, and four 3-angled Silicon photodiodes to avoid reflections between filters and detectors.
The FOV of the PFR is determined by two diaphragms of 3 and 7 mm in diameter, separated by 160 mm, resulting in a FOV of
2.5 degrees. The system is hermetically sealed with a slightly pressurized internal atmosphere (2000 hPa) of dry nitrogen and
temperature stabilized with a Peltier-type thermostat system that maintains the sensor head temperature at $20.0 \pm 0.5°C$ for an
ambient temperature range between -20.0°C and 35.0°C. This system eliminates the need for temperature corrections to the
sensor signal and also prevents accelerated aging of the filters, ensuring the high stability of the PFR (Toledano et al., 2018).
The detectors are only exposed for short periods of time, as an automatic shutter opens every minute for 10 seconds to take
measurements of direct solar radiation, minimizing filter degradation related to exposure. The expected uncertainty in AOD
depends on the uncertainty in the calibration and air mass (Kazadzis et al., 2018a). Wehrli (2000) estimated the uncertainty in
the PFR calibration constant between 0.2% (500 nm) and 1% (368 nm), which leads to an uncertainty in the AOD between
0.002 and 0.01 for a relative air mass equal to 1.

Cloud flagging for PFRs follows some of the basic features that CIMEL uses and also additional ones based on the high (1
minute) measurement frequency of the PFR described in Kazadzis et al. (2018b).

## 140  3  Sites description

### 3.1  Izaña Observatory (IZO)

Izaña observatory (Tenerife, Canary Islands, Spain; 28.309°N, 16.499°W, 2373 m a.s.l.) is a high-mountain subtropical sta-
tion managed by the Izaña Atmospheric Research Center (IARC), belonging to the State Meteorological Agency of Spain
(AEMET). This station predominantly represents the background atmospheric conditions of the subtropical lower troposphere
due to its location above a strong, quasi-permanent layer of thermal inversion resulting from general subsidence processes in
the troposphere (descending branch of the Hadley cell) and the presence of trade winds at lower levels (Carrillo et al., 2016;
Cuevas et al., 2019, 2022; Barreto et al., 2022a). However, the proximity to the Sahara desert introduces an important influence
of mineral dust in its aerosol climatology, making IZO a key location for dust transport monitoring (Rodríguez et al., 2011;
Rodríguez et al., 2015; Barreto et al., 2022a, b).





Aerosol characterization at the site is dominated by free-troposphere conditions throughout the year, with remarkably low aerosol loading ($AOD_{500nm}$ of 0.03 or lower and average Ångström Exponent, $AE_{440-870nm}$, values of 1.01) and a predominant impact of fine-mode aerosols (Barreto et al., 2022b). Dust-laden conditions are predominantly observed in summer, with $AOD_{500nm}$ of 0.15 and $AE_{440-870nm}$ of 0.54 (Barreto et al., 2022b). The predominance of extremely clean atmosphere throughout the year makes IZO a calibration site for GAW-PFR and AERONET-Cimel networks, providing an advantage when

comparing the two instruments since it eliminates to a great extent the errors caused by turbidity or atmospheric instability.

Izaña is an ACTRIS calibration facility, member of ACTRIS/CARS (Center for Aerosol Remote Sensing), responsible for the QA/QC of the automatic sun/sky/lunar photometer measurements.

### 3.2  Valladolid Station

Valladolid station (Valladolid, Castilla Leon, Spain; 41.664°N, 4.706°W, 705 m a.s.l.) is an urban site located in the North-

central part of the Iberian Peninsula. This site is characterized by a continental clean aerosol climatology (Román et al., 2014) with hot and dry summers and cold winters. However, Valladolid can be influenced by both local and regional sources of pollution. The high variability in temperature and humidity leads to a dynamic aerosol environment, with changes in sources and transport pathways over time.

Valladolid is an AERONET calibration center, providing accurate and precise AOD measurements for use in climate and air

quality work. Several studies have shown that Valladolid has relatively low AOD values compared to other urban areas, likely due to the city's location and transport patterns with a monthly mean AOD values ($AOD_{440nm}$) ranging from 0.10 to 0.24 and $AE_{440-870nm}$ from 0.9 to 1.5 (Cachorro et al., 2016). However, the city experiences occasional episodes of elevated AOD associated with transported dust from North Africa and forest fires. For example, Cachorro et al. (2016) found that the highest AOD values in Valladolid occurred during dust episodes from the Sahara desert, with $AOD_{440nm}$ above 0.3 and reaching

values as high as 1-2. These dust events modulate the annual climatology of the aerosols in this area with a total contribution about 11.5% to the total AOD.

Valladolid station is also a member of ACTRIS CARS responsible for the QA/QC of the automatic sun/sky/lunar photometer measurements.

### 4  The Langley Ratio Method

The light traversing the atmosphere suffers attenuation from its interaction with the different atmospheric constituents. This attenuation can be quantified through the Bouguer-Lambert-Beer equation. If we express this equation in terms of the output voltage given by a photometer in a certain spectral band centred at $\lambda$, we have:

$$V_\lambda = \frac{V_{0,\lambda}}{R^2} e^{-\tau_\lambda m},$$ (1)





where $V_{0,\lambda}$ is the calibration coefficient, the photometer output voltage obtained when measuring the sun irradiance at the top

of the atmosphere, $R$ is the Earth-Sun distance normalized to the mean distance, $\tau_\lambda$ is the total optical depth, and $m$ is the

optical airmass. Taking the natural logarithm of both sides of equation 1, we obtain:

$$\ln V_\lambda = \ln \frac{V_{0,\lambda}}{R^2} - m\tau_\lambda, \tag{2}$$

$V_{0,\lambda}$ can be therefore obtained as the extrapolated to zero-air-mass voltage by linear fitting the left-hand side of equation 2

against the optical airmass (usually for air masses ranging between 2 and 5). This is known as the Langley calibration method

and is only applicable when the optical depth is very low and constant. In practice, these very stable atmospheric conditions are

only met in high-altitude stations. Langley calibration technique is an accurate method, with an expected calibration uncertainty

$\sim 0.25$ - 0.5% requiring long observational periods (typically one or two months) to conduct the calibration of reference

(master) instruments (Toledano et al., 2018). Due to the scarcity of places with Langley conditions, the normally expensive

shipping to such remote locations and the longer time period necessary to carry out this calibration, cross-calibration methods

are commonly used in current photometric networks such as AERONET. These methods involve the calibration transference

from some selected master photometers, calibrated through the Langley technique, to the rest of field photometers in more

accessible facilities (Holben et al., 1998; Toledano et al., 2018). In this case, the ratio between the raw direct sun measurements

from the master and the field instrument at the same wavelength is computed for quasi-coincident measurements as the zero-

airmass extrapolated ratio:

$$\frac{V_\lambda^F}{V_\lambda^M} = \frac{V_{0,\lambda}^F}{V_{0,\lambda}^M}, \tag{3}$$

This method is commonly called the Ratio calibration method and is a cross-calibration transference technique comparing

at-ground atmospheric voltage ratios between field and master instruments (normally performed around noon) in the case of

coincident measurements (time difference $< 5$ s). This is a faster calibration method (it usually takes some weeks) requiring

less demanding atmospheric conditions ($\tau_{440nm}$ below 0.15 and cloud-free skies) compared to the Langley technique (Holben

et al., 1998). Uncertainty, in this case, is higher (less than 1% as stated in Holben et al. (1998); Giles et al. (2019)) mainly

because of the higher atmospheric variability and the assumption of similar total optical depth at the time of the coincident

observations. These assumptions and the common Ratio cross-calibration method itself are valid as long as the spectral bands

for both photometers are very similar (i.e. $\Delta\lambda \sim 0$ and similar FWHM). However, this is not always valid, especially when

two photometers measuring at different spectral ranges are compared ($\Delta\lambda$ and FWHM difference are relevant). In this case,

we propose extending the common Ratio technique in equation 3 to include the exponential decrease of each $\tau_\lambda$ in the two

different sets of coincident photometric observations. Therefore, the LR method can be expressed as the ratio of coincident

voltages, as shown in equation 1:

$$\frac{V_{\lambda_F}^F}{V_{\lambda_M}^M} = \frac{V_{0,\lambda_F}^F}{V_{0,\lambda_M}^M} e^{(\tau_{\lambda_M} - \tau_{\lambda_F})m}, \tag{4}$$





where the total optical depth from the field photometer ($\tau_{\lambda_F}$) and its calibration coefficient ($V_{0,\lambda_F}^F$) are a priori not known. $\tau_{\lambda_M}$

can be inferred since the reference instrument is assumed to be calibrated ($V_{0,\lambda_M}^M$) by means of the Langley technique. Taking

the natural logarithms on both sides of equation 4:

$$\ln \frac{V_{\lambda_F}^F}{V_{\lambda_M}^M} = \ln \frac{V_{0,\lambda_F}^F}{V_{0,\lambda_M}^M} + m\Delta\tau, \qquad (5)$$

where $\Delta\tau = \tau_{\lambda_M} - \tau_{\lambda_F}$. This is a critical term in the LR method because it contains the effect of coincident measurements

performed in different spectral bands. Analogous to the standard Langley method, if we assume $\Delta\tau$ to be constant, we obtain

that equation 5 corresponds to the equation of a straight line with slope $m$ and intersection $\ln \frac{V_{0,\lambda_F}^F}{V_{0,\lambda_M}^M}$. Therefore, the linear fit of

the left-hand side of equation 5 with respect to the airmass allows us to obtain the field photometer calibration $V_{0,\lambda_F}^F$ from the

previously obtained master photometer calibration $V_{0,\lambda_M}^M$.

Unlike the standard Langley method, this method is less sensitive to atmospheric variations as $\tau_{\lambda_M}$ and $\tau_{\lambda_F}$ are expected

to vary approximately in the same way. However, the variability of the different atmospheric components can be high enough

to invalidate the assumption of constant $\Delta\tau$, especially as the differences in central wavelengths increase, and for shorter

wavelengths where atmospheric extinction is more significant. This is particularly true for aerosols ($\tau_{\lambda,a}$), as they can exhibit

high variability in concentration and size, leading to significant changes in $\tau_{\lambda,a}$ and Ångström exponent (AE) values and

consequently in $\Delta\tau$. In order to mitigate the effect of such variability, we propose a modification to equation 5 as follows:

$$\ln \frac{V_{\lambda_F}^F}{V_{\lambda_M}^M} - m_r\Delta\tau_r - m_g\Delta\tau_g - m_a\Delta\tau_a = \ln \frac{V_{0,\lambda_F}^F}{V_{0,\lambda_M}^M} + m\Delta\tau', \qquad (6)$$

where $\Delta\tau_r$ represents the difference in Rayleigh optical depth, $\Delta\tau_g$ accounts for the difference in gas optical depth, $\Delta\tau_a$ sym-

bolizes the difference in aerosol optical depth, and $\Delta\tau'$ corresponds to the remaining optical depth difference not adequately

accounted for by the other terms, particularly $\Delta\tau_a$. $\Delta\tau_g$ and $\Delta\tau_r$ can be calculated using the algorithms widely described in the

literature (e.g. Holben et al. (1998), Giles et al. (2019) and Cuevas et al. (2019) and references therein) once the correspond-

ing gas concentration and atmospheric pressure are provided. Specifically, $\Delta\tau_a$ is calculated as $\tau_{\lambda_M,a} - \tau_{\lambda_F,a}$, where $\tau_{\lambda_F,a}$ is

estimated using the Ångström law (Ångström, 1929).

## 5   Results

Coincident photometric measurements obtained from different instruments at various stations, affected by different aerosol

conditions, and processed using various calibration techniques, are compared in this section. We further evaluate the new LR

calibration technique by comparing it to the reference Langley technique for transferring calibration from the GAW-PFR to the

AERONET-Cimel (CE318-TS version) at IZO and Valladolid stations in Section 5.1. Additionally, we assess the effectiveness

of the LR calibration technique when applied to the same instrument, including different Cimel versions (CE318-TS and

CE318-TV12-OC) in Section 5.2, and the same CE318-TS version with slightly different spectral bands in Section 5.3. We





also investigate in this section the use of the LR method for detecting and correcting possible instrumental issues in our photometers.

## 5.1 Calibration transfer from GAW-PFR to AERONET-Cimel

In this section, we present the results of transferring calibration from the GAW-PFR, which is considered the reference instrument for AOD measurements by the WMO, to an AERONET-Cimel (CE318-TS) photometer. Although the GAW-PFR and AERONET-Cimel have distinct characteristics such as different optics, sun tracking systems, relatively different field of view and spectral bands with varying FWHMs and $\lambda_c$, several comparison studies have been carried out in the past to make these different datasets comparable (Kazadzis et al., 2014, 2018a; Cuevas et al., 2019, among others). For example, Cuevas et al. (2019) conducted an exhaustive, long-term comparison in terms of AOD, which showed excellent traceability of AERONET-Cimel AOD with the GAW-PFR AOD reference at 440, 500, and 870 nm channels, with poorer traceability results in the UV range. These authors also showed the important effect that the different FOV between the two photometers has on the AOD monitoring, with an important AOD underestimation in the case of PFR under the presence of coarse particles due to the enhanced forward aerosol scattering, specially at the shorter wavelengths.

Our study was conducted at two different locations, IZO and Valladolid, where co-located PFR and Cimel were used to collect measurements almost simultaneously (within 1 minute) over a period of six months. The time period encompassed measurements with the AERONET photometer #1089 between 1 July - 31 December 2021 in the case of Izaña, and with the AERONET photometer #904 between 1 July - 31 December 2022 in the case of Valladolid. The different aerosol regimes at these two sites allowed us to assess the new LR calibration method under different atmospheric conditions: one pristine high-altitude site, suitable for performing the Langley calibration, and an urban site with a moderate impact of aerosol variability, used in AERONET as a cross-calibration station. The six-month period was chosen because it is the common recalibration interval of reference instruments in GAW-PFR and AERONET-Cimel networks (Kazadzis et al., 2018b; Giles et al., 2019). According to Section 2, these instruments have been quality assured including two independent cloud screening (GAW-PFR and AERONET-Cimel level 2.0 algorithms).

Daily $V_{0,\lambda}$ values retrieved with the LR method ($V_{0,\lambda}^{LR}$) over the six-month period at the two sites were computed using the ratio (expressed as in equation 6) of coincident PFR ($V_{\lambda_M}^{M}$) and Cimel ($V_{\lambda_F}^{F}$) measurements (i.e., $V_{\lambda_F}^{F}/V_{\lambda_M}^{M}$) for an airmass range from 2 to 5, with two daily $V_{0,\lambda}^{LR}$ values associated with the morning and afternoon branches. We have considered the closest spectral bands between the two photometers, i.e., 1020/862, 1640/862, 870/862, 675/500, 440/412, 500/500, 1020i/862, 380/368 and 340/368. It means a $\Delta\lambda$ between instruments (Cimel versus PFR) ranging from 778 nm (for Cimel 1640 nm spectral band) down to zero at 500 nm . Cimel exo-atmospheric output voltage obtained by using the Standard Langley method $V_{0,\lambda}^{SL}$ for the #1089 and #904 Cimel photometers at IZO were used as a reference. The two $V_{0,\lambda}^{SL}$ values, one per photometer, were considered valid for the six-month period of measurements used in the present study. Figures 1 and 2 show the daily relative differences, in %, between $V_{0,\lambda}^{SL}$ and $V_{0,\lambda}^{LR}$ for the different CE318 spectral bands (except 937 nm) at the two sites, respectively.



**Figure 1.** Daily $V_{0,\lambda}$ relative differences (in %) between the calibration constant obtained by applying the standard Langley calibration ($V_{0,\lambda}^{SL}$) to the CE318-TS instrument #1089 at Izaña and the calibration constant transferred to the Cimel from the PFR applying the Langley Ratio method ($V_{0,\lambda}^{LR}$) to daily observations at Izaña. Each figure corresponds to the nine spectral bands of the CE318-TS, compared to the nearest PFR spectral band by means of the ratios ($V_{\lambda_F}^{F}/V_{\lambda_M}^{M}$) at 1020/862, 1640/862, 870/862, 675/500, 440/412, 500/500, 1020i/862, 380/368 and 340/368. The color indicates the standard deviation of $\Delta\tau$.

At IZO, Figure 1 and Table 1 show that, on average, the relative differences in the calibration constant retrieved by the Standard Langley technique and the LR ($\overline{\Delta V_0}$) and their standard deviation ($\sigma(\Delta V_0)$) are generally low for spectral bands outside the UV range, below 0.45 % and 0.77 %, respectively. The high $\sigma(\Delta V_0)$ values observed at the 1640 nm spectral band (0.77 %) are attributed to the significant $\Delta\lambda$ used in the LR, which employs the photometric information obtained by the master PFR at the spectral band centred at 862 nm to calibrate this Cimel spectral band in the NIR ($\Delta\lambda$ of 778 nm).



**Figure 2.** Daily $V_{0,\lambda}$ relative differences (in %) between the calibration constant obtained by applying the standard Langley calibration ($V_{0,\lambda}^{SL}$) to the CE318-TS instrument #904 at IZO and the calibration constant transferred to the Cimel from the PFR applying the Langley Ratio method ($V_{0,\lambda}^{LR}$) to daily observations at Valladolid. Each figure corresponds to the nine spectral bands of the CE318-TS, compared to the nearest PFR spectral band by means of the ratios ($V_{\lambda_F}^{F}/V_{\lambda_M}^{M}$) at 1020/862, 1640/862, 870/862, 675/500, 440/412, 500/500, 1020i/862, 380/368 and 340/368. The color indicates the standard deviation of $\Delta\tau$

In the UV range, these values are found to be higher, up to 0.78 % for $\overline{\Delta V_0}$ and 1.41 % for $\sigma(\Delta V_0)$. It is well-known that there are higher uncertainties and temperature dependence in the Cimel photometers in this spectral range (Giles et al., 2019). Although the CE318-TS signal is typically corrected in AERONET for temperature across most wavelengths using standard integrating spheres, such temperature calibration is not performed for the UV filters due to limited source power in this spectral range. These results are also consistent with the expected increase in variation in the $\Delta\tau$ term due to the more significant






**Table 1.** Daily mean relative differences and standard deviation in $V_{0,\lambda}$ (in %) between the calibration constant obtained by applying the standard Langley calibration ($V_{0,\lambda}^{SL}$) to the reference GAW-PFR and the calibration constant transferred from the GAW-PFR to the AERONET-Cimel applying the Langley Ratio method ($V_{0,\lambda}^{LR}$) to daily observations at IZO for the different CE318-TS spectral bands. Three different time periods have been included: the whole period (all data), the period corresponding between July to September 2022 (summertime period), denoted as $1^{st}$ period, and the $2^{nd}$ period, from October to December.

| Cimel spectral Band (nm) | 1640 | 1020 | 870 | 675 | 500 | 440 | 1020i | 380 | 340 |
|---|---|---|---|---|---|---|---|---|---|
| $\overline{\Delta V_0}$ (all data) | -0.21 | 0.06 | 0.10 | 0.27 | 0.44 | 0.34 | -0.06 | 0.78 | 0.67 |
| $\sigma(\Delta V_0)$ (all data) | 0.77 | 0.42 | 0.36 | 0.42 | 0.57 | 0.58 | 0.40 | 1.03 | 1.41 |
| $\overline{\Delta V_0}$ ($1^{st}$ period) | -0.21 | 0.19 | 0.17 | 0.39 | 0.57 | 0.42 | 0.01 | 0.72 | 0.12 |
| $\sigma(\Delta V_0)$ ($1^{st}$ period) | 0.98 | 0.50 | 0.44 | 0.51 | 0.62 | 0.68 | 0.48 | 1.13 | 1.55 |
| $\overline{\Delta V_0}$ ($2^{nd}$ period) | -0.20 | -0.07 | 0.03 | 0.15 | 0.29 | 0.24 | -0.13 | 0.84 | -0.10 |
| $\sigma(\Delta V_0)$ ($2^{nd}$ period) | 0.45 | 0.25 | 0.20 | 0.24 | 0.46 | 0.44 | 0.28 | 0.90 | 1.25 |

aerosol extinction in this spectral range. Looking at Figure 1, two different time periods affecting the $V_0$ differences can be observed, with more scatter and higher $\sigma(\Delta\tau)$ in the first period between July and September 2022. This period corresponds to the frequent dust outbreaks affecting Izaña during summertime (Barreto et al., 2022a, b), and therefore, more effect of dust forward scattering in AOD retrieval uncertainty due to the different instrument FOVs (Cuevas et al., 2019). The impact of

circumsolar irradiance on the higher FOV of the PFR radiometer was already studied by Cuevas et al. (2019), who found an underestimation of AOD by PFR, which is more significant at lower wavelengths, depending on dust particle radius and AOD load and is considered non-negligible for $AOD_{500nm}$ values $> 0.1$. This effect will directly translate into the $\Delta\tau'$ term in the slope of the LR analysis (Eq. 6).

To investigate the performance of the LR for different aerosol regimes, we included two different time periods in Table 1: the

period from July to September 2022 (a summertime period with a significant amount of dust), denoted as the $1^{st}$ period, and the $2^{nd}$ period, from October to December, which was expected to have clean conditions at the site. Mean $AOD_{500nm}$ values of 0.12 (with a standard deviation of 0.09) and 0.03 (with a standard deviation of 0.03) were retrieved for these two periods, respectively. We found considerably lower $\overline{\Delta V_0}$ and $\sigma(\Delta V_0)$ values in the $2^{nd}$ period when the influence of mineral dust and the possible impact of the different FOV on the LR method through the term $\Delta\tau$ were minimized. Under these non-dusty

conditions, we observed $\overline{\Delta V_0}$ values between -0.20 and 0.30 % and $\sigma(\Delta V_0)$ values up to 0.46 % for spectral bands outside the UV range. In the UV range, $\overline{\Delta V_0}$ and $\sigma(\Delta V_0)$ values reached up to 0.84 and 1.25%, respectively, for the $2^{nd}$ period.

In the case of the Valladolid station, Figure 2 and Table 2 depict the time evolution of $\Delta V_0$ and their mean and standard deviation values. A lack of $\Delta V_0$ seasonal dependence was found at this station, indicating a lower impact of aerosol seasonality at Valladolid in comparison to Izaña high-mountain station. However, some periods with higher $\Delta V_0$ values associated with

higher values of $\sigma(\Delta V_0)$ can be observed in Figure 2, especially at longer wavelength spectral bands. These periods, observed in July and August and not related to the increase in AOD conditions, may be caused by instrumental issues such as calibration





**Table 2.** Daily mean relative differences and standard deviation in $V_{0,\lambda}$ (in %) between the calibration constant obtained by applying the standard Langley calibration ($V_{0,\lambda}^{SL}$) to the reference GAW-PFR and the calibration constant transferred from the GAW-PFR to the Cimel applying the Langley Ratio method ($V_{0,\lambda}^{LR}$) to daily observations at Valladolid for the different CE318-TS spectral bands.

| Cimel spectral Band (nm) | 1640 | 1020 | 870 | 675 | 500 | 440 | 1020i | 380 | 340 |
|---|---|---|---|---|---|---|---|---|---|
| $\overline{\Delta V_0}$ | 0.38 | 0.68 | 0.45 | 0.93 | 1.04 | 1.61 | 0.05 | 1.36 | 1.44 |
| $\sigma(\Delta V_0)$ | 0.93 | 0.57 | 0.50 | 0.67 | 0.54 | 1.10 | 0.54 | 1.23 | 1.83 |

problems or dirtiness in one of the photometers. Additionally, an important bias in the $\Delta V_0$ values in the Cimel 675, 500, and 440 nm spectral bands (Figures 2(d), (e), and (f)) was also observed, up to 1.61 %. This difference could also be due to possible instrumental problems. $\overline{\Delta V_0}$ values up to 0.68 % were found in the longer wavelengths (1640-870nm), with $\sigma(\Delta V_0)$ ranging

from 0.50 to 0.93 % in 1640 nm due to the high wavelength difference between spectral bands involved in the LR method. Similarly to Izaña, comparatively higher differences and standard deviations were found in the UV range (up to 1.44 and 1.83 %, respectively).

## 5.2    Langley Ratio application between different CE318-T photometers

One important application of the LR method is its use in calibrating different versions of the CE318-T when the spectral

bands between them are different. This is the case for the CE318-TS and CE318-TV12-OC photometers, which are standard instruments in AERONET and AERONET-OC, respectively. As in the previous section, each of the 12 spectral bands of the CE318-TV12-OC has been compared to the nearest CE318-TS spectral band using the ratios $V_{\lambda_F}^F/V_{\lambda_M}^M$. This involves a comparison between 1020/1020, 865/870, 779/870, 665/675, 620/675, 560/500, 510/500, 1020i/1020i, 490/500, 442/440, 412/440, 400/440, with a $\Delta\lambda$ up to 91 nm (at 779 nm). These two instruments were used to measure at Izaña for a period of 20 days,

from January 09 to January 29, 2023. The reference calibration value in this case is the standard Langley calibration ($V_{0,\lambda}^{SL}$) of the CE318-TS photometer, which is compared to the daily $V_{0,\lambda}^{LR}$ values in Figure 3 and Table 3. The temporal evolution of the $V_{0,\lambda}$ relative differences in % has been split into the morning (blue circles) and afternoon (orange circles). With the exception of 1020 (Figure 3(a)), low $\overline{\Delta V_0}$ and $\sigma(\Delta V_0)$ values were obtained over the time period. The considerably higher differences in the 1020 nm spectral band measured with the Silicon detector are related to the considerably important temperature dependence

of the Silicon detectors needed at this wavelength range (Holben et al., 1998) and the lack of temperature characterization of the CE318-TV12-OC photometers. In the rest of the spectral bands, $\overline{\Delta V_0}$ ranged between 0.17 and 0.69 in the morning and between 0.02 and 1.13 during the afternoon, with minimum differences in the longest wavelength channel (1020i nm) and maximum in the shortest wavelength channel (400 nm). A similar behaviour was observed for $\sigma(\Delta V_0)$, with remarkably lower values ranging from 0.01 to 0.21 during the morning and slightly higher (0.04-0.19) in the afternoon excluding 1020 filter. The

difference between morning and afternoon in terms of variability is attributed to the effect of atmospheric turbulence during the afternoon period, while the mean differences observed at shorter wavelengths are attributed to the joint effect of Cimel





**Table 3.** Daily mean relative differences and standard deviation in $V_{0,\lambda}$ (in %) between the calibration constant obtained by applying the standard Langley calibration ($V_{0,\lambda}^{SL}$) to the reference CE318-TS and the calibration constant transferred to the CE318-TV12-OC from the CE318-TS applying the Langley Ratio method ($V_{0,\lambda}^{LR}$) to daily observations at Izaña for the different CE318-TV12-OC spectral bands.

| Cimel Spectral Band (nm) | 1020 | 865 | 779 | 665 | 620 | 560 | 1020i | 510 | 490 | 442 | 412 | 400 |
|---|---|---|---|---|---|---|---|---|---|---|---|---|
| $\overline{\Delta V_0}$ (am) | -0.78 | 0.25 | 0.24 | 0.31 | 0.37 | 0.47 | 0.17 | 0.35 | 0.34 | 0.48 | 0.53 | 0.69 |
| $\sigma(\Delta V_0)$ (am) | 0.56 | 0.01 | 0.03 | 0.05 | 0.07 | 0.06 | 0.21 | 0.04 | 0.04 | 0.07 | 0.08 | 0.11 |
| $\overline{\Delta V_0}$ (pm) | -0.78 | 0.26 | 0.22 | 0.30 | 0.34 | 0.43 | 0.02 | 0.37 | 0.42 | 0.70 | 0.83 | 1.13 |
| $\sigma(\Delta V_0)$ (pm) | 0.56 | 0.05 | 0.07 | 0.07 | 0.04 | 0.14 | 0.16 | 0.10 | 0.12 | 0.14 | 0.16 | 0.19 |

tracking (no pointing refinement in the last measurements performed at shorter wavelengths) and optical differences between instruments.

### 5.3 Application of the LR method to detect and correct instrumental problems

In this section, we demonstrate the application of the Langley Ratio method to detect and correct possible instrumental issues in our photometers. In Sections 5.1 and 5.2, we demonstrated the suitability of the LR method to transfer the calibration between instruments with different spectral bands. However, we have not checked the calibration error that the use of the conventional Ratio cross-calibration approach in these $\Delta\lambda$ conditions can imply. We also observed the impact of different FOV between instruments and possible instrumental issues such as calibration, temperature dependence in some spectral bands (1020 nm
measured with the Silicon detector and UV filters), as well as the joint effect of the lack of refinement during the Cimel tracking and the possible optics difference between similar spectral bands.

#### 5.3.1 Error in the Ratio calibration method and tracking errors in the UV

The current analysis has focused on the UV spectral region, which is affected by higher errors, and has been performed using two Cimel photometers of the same version (CE318-TS) to rule out any FOV impact. It is precisely in this spectral band that
the errors are higher due to higher extinction, temperature dependence, and tracking problems. Since these photometers have the same optics and spectral bands, the calibration could theoretically be transferred from the master to the field instrument by applying the common Ratio cross-calibration method. In this sense, two AERONET-Cimel photometers, #904 and #942, have been selected due to the relatively high $\Delta\lambda$ difference in the 340 nm spectral band. $\lambda_{c,340nm}$ is 340.92 nm in the case of photometer #904 and 339.48 nm in the case of #942, yielding a $\Delta\lambda$ of 1.44 nm.

Figure 4 shows the relative difference between the calibration made by the standard Langley method ($V_{0,\lambda}^{SL}$), performed just before the analysis period, and the calibration retrieved through the common Ratio cross-calibration method ($V_{0,\lambda}^{R}$) without any temperature correction, shown as green crosses. We have also included in this analysis the comparison between the standard Langley and the Langley Ratio method ($V_{0,\lambda}^{LR}$) without any temperature correction, shown as blue and orange crosses. This analysis was performed in Valladolid for the same six-month period presented in Section 5.1, and it was also split into morning





and afternoon data. In the plot, we can observe the presence of a significant bias (-1.6 %) at the beginning of the period between the standard Langley and Ratio calibration techniques. This difference followed a seasonal dependence due to the presence of an appreciable $\Delta\lambda$ that was not included in the analysis, but was included in the LR method through the airmass term in Eq. 4. The seasonal difference is maximum (-3.8 %) at the end of December. The performance of the LR method in Figure 4 shows relatively low $\overline{\Delta V_0}$ and $\sigma(\Delta V_0)$ values with no seasonal dependence. Morning/afternoon differences are also observed in this

figure, with $\overline{\Delta V_0}$ ($\sigma(\Delta V_0)$) values of 0.16 % (0.66) for morning data and -0.72 % (0.66) during the afternoon.

We have delved into the origin of these daily differences by improving the solar tracking of two test CE318-TS photometers at Izaña. This was done over a sequence of six days (26-31 May 2021) to correct the pointing before each spectral measurement. This analysis has been performed in terms of ratio of voltages ($V_F/V_M$) with these two test instruments operating with a new firmware version which allow them to perform a sequence of triplet measurements with the pointing correction followed by

a sequence without this correction. This correction compensates for the Earth's rotational movement before each spectral Cimel measurement. A significant improvement is observed in Figure 5 for the 340 nm spectral band on one of the six days analyzed (orange circles represent corrected data, and blue circles represent uncorrected data). This figure demonstrates a significant reduction in the morning/afternoon dependence of the ratios (ratio asymmetry) after the implementation of the pointing refinement. The average ratios for morning (afternoon) were 1.0925 (1.0941) and 1.0961 (1.0948) before the change

in tracking and 1.0961 and 1.0948 after the change during the 6-day period. This translates to a percentage difference in morning/afternoon with average ratios of 0.33 % without correction and 0.06 % after correction. We have observed that the asymmetry between morning and afternoon measurements at shorter wavelengths (those measured at the end of the Cimel measurement routine) can be reduced by performing the tracking correction. This asymmetry depends on the instruments involved, and we have observed instruments with a ratio asymmetry of up to 2 %.

**5.3.2 Temperature dependence on UV filters**

Another source of problems related to the Cimel photometers is the temperature dependence on UV spectral bands that cannot be corrected with the current resources used in AERONET. To address this issue, we have conducted an analysis using the LR technique to estimate the influence of temperature on UV signals. Specifically, we have used two photometers at Izaña: #904 and #942, the two photometers involved in the analysis displayed in Figure 4 of the comparison performed at Valladolid.

We selected photometric measurements at this high-altitude site in a four-month period to transfer the calibration from a temperature-corrected CE318-TS filter (440 nm) to the UV filters (340 and 380 nm) using the LR technique. Figure 6 shows that $\Delta V_0$ varies linearly with the temperature, with an important decay in temperature of $\sim 0.09\mathrm{x}10^{-2}/°$ in the case of 380 nm and $\sim 0.03\mathrm{x}10^{-2}/°$ in the case of 340 nm. This equation can be used to roughly correct the temperature effect in these two spectral bands. We have verified this temperature correction by looking at the daily $V_{0,\lambda}$ relative differences displayed in

Figure 4 (closed circles). The new $V_{0,\lambda}$ relative differences have notably decreased in terms of $\overline{\Delta V_0}$ and $\sigma(\Delta V_0)$ during the morning, with a reduction of 31 % and 14 %, respectively. In the case of the LRs during the afternoon the reduction is only observed in the dispersion, with $\sigma(\Delta V_0)$ -4.5 % lower.





# 6   Conclusions

This paper applies sun-photometer synergies to improve calibration transference between different sun-photometers and also
enhance their quality assurance and quality control. Coincident PFR and CE318-T photometric observations from Izaña (GAW
core) and Valladolid (associated GAW) stations, over a six-month period, were used to analyze the performance of the new Lan-
gley Ratio (LR) calibration method. This new method has been proven to be effective in transferring the calibration from ref-
erence instruments, which were calibrated using Langley calibration at Izaña, to field instruments under different atmospheric
conditions. Photometric information from Izaña high-mountain and Valladolid urban stations, Langley and cross-calibration
sites in AERONET, respectively, revealed the presence of important external factors affecting the LR method, such as the dif-
ferent FOV between photometers in dusty conditions at Izaña, or the probable presence of some calibration issues in Valladolid.
Not considering these external factors, our results showed very low average $V_{0,\lambda}$s between Langley and LR, up to 0.29%, with
standard deviations of these $V_{0,\lambda}$s relative differences of up to 0.46% for Izaña. In the case of Valladolid, these average differ-
ences are up to 1.04 %, with standard deviations between 0.50 and 0.93 % (in the case of the 1640 nm spectral band). Higher
differences were observed in the shorter wavelengths (up to 0.84 % in Izaña and 1.61 % in Valladolid), attributed to the higher
errors affecting Cimel spectral bands (higher Rayleigh contribution, temperature dependence on filters, and tracking issues).

A subsequent calibration transfer between the two reference instruments in AERONET and AERONET-OC (CE318-TS and
CE318-TV12-OC) led us to conclude that the LR method is valid in the case of a $\Delta\lambda$ up to 91 nm, with relative $V_{0,\lambda}$ between
Langley and LR ranging from 0.17 % to 0.69 % for LR calibrations performed in the morning and from 0.02 % to 1.13 % in
the afternoon. Remarkably low standard deviations of $V_{0,\lambda}$ (0.01-0.21) were found during morning LRs, with higher dispersion
(0.04-0.19) observed in the afternoon. Atmospheric turbulence is expected to be the cause of the difference between morning
and afternoon LRs in terms of variability, while the joint effect of solar tracking and different optics between instruments is
expected to be the reason for the high mean differences observed at shorter wavelengths. In this case, the results associated
with the CE318-TV12-OC spectral band are worse due to the lack of temperature correction in the reference AERONET-OC
instruments.

Our results revealed that the use of the common Ratio cross-calibration technique in the case of two instruments with a
significant $\Delta\lambda$ between spectral bands can lead to calibration errors of up to -3.8 % in the UV spectral bands, and this error
has a seasonal dependence due to not considering the exponential term of the equation 4.

Finally, the LR method has been demonstrated to be a useful tool for detecting and correcting possible instrumental issues
in our photometers. It was used to detect and estimate the effect of Cimel solar tracking and different optics on photometric
Cimel information throughout the day (morning versus afternoon data). Average $V_{0,\lambda}$ differences of 0.16 % and -0.72 % were
found for the Langley-LR comparison during the morning and afternoon, respectively. Regarding the temperature dependence
on UV spectral bands, the LR method was used to roughly correct the temperature effect in these two spectral bands. Our
results revealed an important decay in temperature of $\sim 0.09 \times 10^{-2}/°$ in the case of 380 nm and $\sim 0.03 \times 10^{-2}/°$ in the case of
340 nm. This equation can be used to implement, for the first time, a temperature correction in these two spectral bands.





In conclusion, this hybrid calibration technique between the Langley plot reference method and the faster and less accurate Ratio cross-calibration method appears to be a suitable technique for transferring the calibration between instruments with different spectral bands. However, its validity relies on the aerosol variation during the calibration period, although it is less sensitive to aerosol variations compared to the classical Langley calibration method.



**Figure 3.** Daily $V_{0,\lambda}$ relative differences (in %) between the calibration constant obtained by applying the standard Langley calibration ($V_{0,\lambda}^{SL}$) to the CE318-TV12-OC at Izaña and the calibration constant transferred to the CE318-TV12-OC from the CE318-TS applying the Langley Ratio method ($V_{0,\lambda}^{LR}$) to daily observations at Izaña. Each figure corresponds to the 12 spectral bands of the CE318-TV12-OC, compared to the nearest CE318-TS spectral band by means of the ratios ($V_{\lambda_F}^{F}/V_{\lambda_M}^{M}$) at 1020/1020, 865/870, 779/870, 665/675, 620/675, 560/500, 510/500, 1020i/1020i, 490/500, 442/440, 412/440, 400/440. The orange circles represent relative differences in the morning and the blue circles in the afternoon.





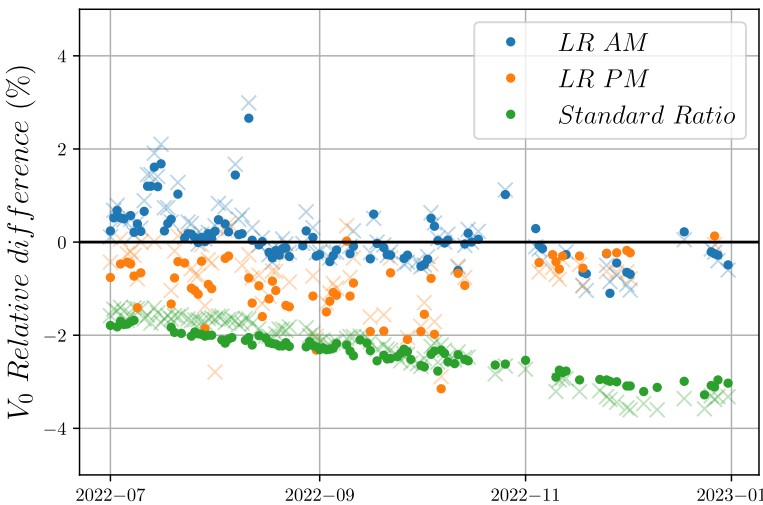

**Figure 4.** Daily $V_0$ relative differences (in %) between the calibration constant obtained by applying the standard Langley calibration ($V_0^{SL}$) using a CE318-TS at Izaña and the calibration constant transferred between two CE318-TS using the Ratio method ($V_0^R$) and the LR method ($V_0^{LR}$) to daily observations at Izaña for 340 nm. Crosses indicate those results without temperature correction and circles once temperature correction was applied to this filter. Green crosses/circles represent relative differences between $V_0^{SL}$ and $V_0^R$, blue crosses/circles represent relative differences between $V_0^{SL}$ and $V_0^{LR}$ in the morning and orange crosses/circles in the afternoon.

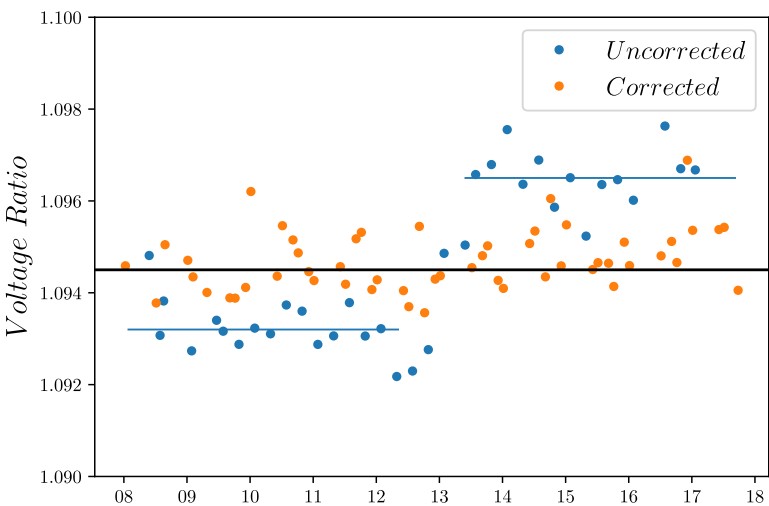

**Figure 5.** Voltage Ratios ($V_F/V_M$) of two instruments measuring one day (31 May 2021) at Izaña at the 340 nm spectral band with the tracking correction (corrected, in orange) and without the tracking correction (uncorrected, in blue). The black horizontal line represents the mean value of corrected ratios. The two blue horizontal lines represent the average ratios during the morning and the afternoon.



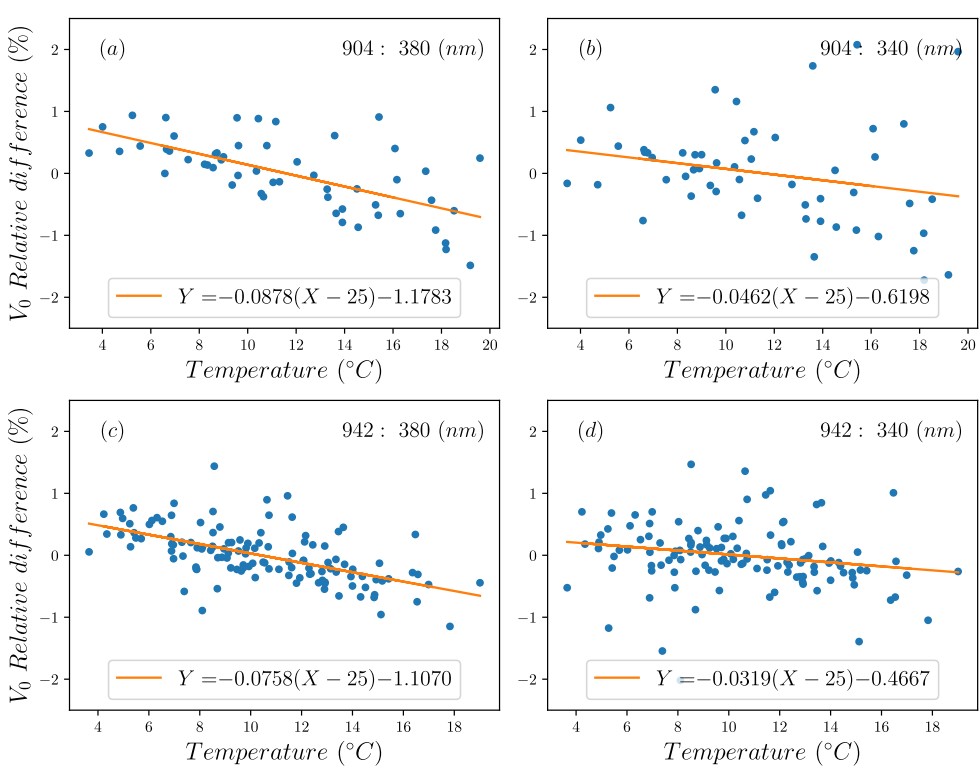

**Figure 6.** Relative $V_0$ difference between the daily $V_0^{LR}$ and the average $V_0^{LR}$ value in a four-month period (March - June, 2021 for #904 and January - April, 2021 for #942) against temperature in Izaña for two instruments at the two UV spectral bands: (a) for #904 at 380 nm, (b) for #904 at 340 nm, (c) for #942 at 380 nm and for (d) #942 at 340 nm.





*Data availability.* The data from the CE318-T photometers used in the present study for Izaña and Valladolid stations can be provided by request to the corresponding author Antonio F. Almansa at f-almansa@cimel.fr. GAW-PFR data can be provided by request to PMOD/WRC staff at Davos (natalia.kouremeti@pmodwrc.ch).

*Author contributions.* A.A., A.B., N.K., R.G. and C.T. designed and wrote the structure and methodology of the paper. A.A. performed the calculations required for this analysis. J.G., R.D.G., Y.G., S.K. and S.V. discussed the results and participated in the retrievals analysis. N.K.
and A.M. performed the calibration of PFR photometers. O.A. and V.C. performed the maintenance and daily checks of the instrumentation at Izaña. E.C. and V.E.C. ensured the provision of funds for the aerosol measurement programme at Izaña and Valladolid, respectively. All authors discussed the results and contributed to the final paper.

*Competing interests.* The authors declare that they have not conflict of interest.

*Acknowledgements.* This work has been developed within the framework of the activities of the World Meteorological Organization (WMO)
Commission for Instruments and Methods of Observations (CIMO) Izaña Testbed for Aerosols and Water Vapour Remote Sensing Instruments. AERONET sun photometers at Izaña were calibrated through the AEROSPAIN Central Facility (https://aerospain.aemet.es/) supported by the European Community Research Infrastructure Action under the ACTRIS grant, agreement no. 871115. This work has been supported by the European Metrology Program for Innovation and Research (EMPIR) within the joint research project EMPIR 19ENV04 MAPP. The EMPIR is jointly funded by the EMPIR participating countries within EURAMET and the European Union. The authors also
acknowledge the support from the Ministerio de Economía y Competitividad from Spain through the project SYNERA (PID2020-521-118793GA-I00), Ministerio de Ciencia e Innovación (grant no. PID2021-127588OB-I00), the Junta de Castilla y León (grant no. VA227P20) as well as to IZO and Valladolid staff for maintaining the instrumentation, thus ensuring the quality of the data.





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
