# Peer review of "The Langley Ratio method, a new approach for transferring photometer calibration from direct sun measurements"

_Atmospheric Measurement Techniques, 2023_

## Author Comment (AC1)

**Manuscript: Preprint amt-2023-108**

**Title: The Langley Ratio method, a new approach for transferring photometer calibration from direct sun measurements**

**Response to Referee#3**

The authors appreciate the overall positive response of the Referee #3 and we would like to thank for his/her constructive comments. In the following, the Referee suggestions (in bold) are in detail addressed (the author's responses are provided below in blue colour).

**The authors present a new method for transferring calibration from a reference photometer, using a synergetic approach when master and field instruments have different spectral bands. This new method, so called Langley Ratio method, was first applied between a PFR and a CE318-TS photometer, because these two photometers have different optics, sun-tracking systems and spectral bands. The campaign and validation at Izaña Observatory (IZO) and Valladolid showed the very low relative differences and standard deviations in the calibration constant transferred in Izaña from PFR to Cimel, up to 0.29 % and 0.46 %. This is really a satisfactory result, and the following studies vitrificated that the Langley Ratio method is a robust and suitable tool for transferring calibrations, detecting and correcting possible instrumental issues.**

**In summary, the paper is well-written and logically organized. I think it provided a useful way to conduct the calibration of sun photometer in a more efficient pattern, which is important thing for the observation network all over the world. So, I recommend this paper to be published in AMT after revision, but I still have some question that the authors should take into consideration as below:**

**1, Line 49. I think it is unnecessary to emphasize the Valladolid site is not a part of GAW.**

We agree with this comment. This information will be deleted from the manuscript.

**2, Line 66. The authors should explain more about "SI", as we don't know what is the "SI".**

We missed the acronym definition. SI is for International System of Units. We will add this information in the text.

**3, Line 70. I think the LR method is useful for the sun photometers in different/same spectral bands. However, the sentence here implies it just for the different spectral bands. The authors should check this.**

Line 70 states that "In this paper, we present a new methodology specifically designed to be applied when the calibration transference is carried out between two photometers with different spectral bands in terms of central wavelength ($\lambda_c$) or Full-Width-at-Half-Maximum (FWHM)." The authors think that LR method is a hybrid calibration technique (between the Langley plot reference method and the faster and less accurate

Ratio cross-calibration method) suitable for transferring the calibration between instruments with different spectral bands. In the case of the calibration transference between instrument with similar spectral bands and coincident measurements LR method converges to the Ratio cross-calibration method.

**4, Line 90. I think a table should be more useful here, to highlight the different spectral bands of device in this paper. So that to avoid much too long title in your Figures, repeating the bands.**

We agree with this referee's comment, which coincides with the Referee's 2 comment #6. We will add two tables, one for section 2.1 and another one for section 2.2 describing the CE318-T and the PFR photometers:

Table 1: Main features of the CE318-TS and CE318-TV12-OC sun photometers used in this study.

|  | CE318-TS | CE318-TV12-OC |
|---|---|---|
| Type of instrument | Standard version, Reference instrument in AERONET | Reference instrument in AERONET-OC (Ocean Color) |
| Type of observation | Automatic sun–sky tracking | Automatic sun–sky–sea tracking |
| Available standard channels | 340, 380, 440, 500, 675 nm, 870, 1020, 1640 nm | 400, 412.5, 442.5, 490, 510, 560, 620, 665, 779, 865, 937, and 1020 nm |
| FWHM | 2 nm (340 nm), 4 nm (380 nm), 10 nm (VIS-NIR), 25 nm (1640 nm) | 10 nm |

Table 2: Main features of the CE318-TS and the GAW-PFR sun photometers used in this study

|  | CE318-TS | PFR |
|---|---|---|
| Type of instrument | Standard version, Reference instrument in AERONET | Standard version, Reference instrument in GAW-PFR |
| Type of observation | Automatic sun–sky tracking | Automatic continuous direct sun irradiance |
| Available standard channels | 340, 380, 440, 500, 675 nm, 870, 1020, 1640 nm | 368, 412, 500, 862 nm |
| FWHM | 2 nm (340 nm), 4 nm (380 nm), 10 nm (VIS-NIR), 25 nm (1640 nm) | 5 nm |
| FOV | 1.3° | 2.5° |
| Sun tracker | Robot specifically designed by CIMEL and controlled in conjunction with the radiometer | Any sun tracker with a resolution of at least 0.08° |

**5, Line 108. Maybe "every 15 minutes (in default)" is more accurate. This is an adjustable option in control box of CE318.**

We agree with the referee comment, we will change to "every five minutes (this is the default value in the last firmware version, but it can be adjusted between 2 and 15 minutes)"

**6, Line 195. The variable in equation 2 is undefined, please check.**

We agree with the referee comment (assuming that the referee refers to equation 3 not 2), which also coincides with Referee's 2 comment #11. We have changed the text from line 188 as follows:

Due to the scarcity of locations with Langley conditions, the typically high costs associated with shipping equipment to such remote areas, and the extended time required to conduct this calibration, alternative methods have been developed (Soufflet et al. 1992; Schmid et al. 1998; Holben et al. 1998; Fargion et al. 2001). Specifically, transferring calibration from a Langley-calibrated reference instrument ($V_{0,\lambda}^{M}$), referred to as the "master," to uncalibrated instruments ($V_{0,\lambda}^{F}$), known as "field" instruments, conducted in more accessible facilities offers a practical solution for calibrating multiple instruments simultaneously. In this regard, AERONET applies the method exposed by Fargion et al. 2001, where the calibration of the field instrument, $V_{0,\lambda}^{F}$, is determined by calculating the ratio between equation 1 applied to the field instrument and equation 1 applied to the master instrument for measurements that are both coincident in time and within the same spectral band. Consequently, this ratio can be expressed in terms of quasi-coincident ratios between raw direct sun measurements from the master ($V_{\lambda}^{M}$) and the field instrument ($V_{\lambda}^{F}$) as follows:

$$V_{0,\lambda}^{F} = \frac{V_{\lambda}^{F}}{V_{\lambda}^{M}} \cdot V_{0,\lambda}^{M},$$

**7, Line 416. In conclusion part, the authors should give us some advice that the shortage of LR ratio method, or the un-suitable case, to avoid the calibration uncertainty.**

As stated in Section 4, where the method LR is described, $\Delta\tau$ is the critical term in the LR formulation. The validity of the LR method relies on the fact that $\Delta\tau$ is assumed to be constant. Despite being less sensitive to atmospheric variations than the standard Langley method, this assumption can be compromised in cases of high atmospheric extinction and high variability in aerosol concentration and size, leading to significant changes in $\tau_{\lambda,a}$ and Ångström exponent (AE).

To properly address this question, we have decided to conduct a sensitivity study of $V_0$ obtained with the LR method concerning the variability in AOD and AE. To do this, we have modified the sensitivity study of $\Delta\tau_a$ concerning AOD and AE, which was conducted to respond to a question from the Referee 1. The details of this study are presented below.

First, we created a set of synthetic measurements by applying the Bouguer-Lambert-Beer equation (equation 1 of the preprint article) for a range of $\tau$ values, both for the master

photometer with CWL $\lambda_M$ and for the field photometer with CWL $\lambda_F$. To generate this set of synthetic measurements, we assumed that the contribution from Rayleigh scattering and gas absorption remains constant, while the contribution due to aerosols varies. Thus, for the master aerosol contribution, we considered a range of AOD values ($\tau_{\lambda_M,a}$) randomly distributed in a normal distribution characterized by their mean and standard deviation. For the field instrument, the AOD was calculated from the master instrument using the Ångström law, that is:

$$\tau_{\lambda_F,a} = \tau_{\lambda_M,a} \left(\frac{\lambda_F}{\lambda_M}\right)^{-\alpha} \tag{1}$$

where $\alpha$ is the Ångström exponent. In this case, we have also considered a range of $\alpha$ values randomly distributed in a normal distribution characterized by their mean and standard deviation.

Once these synthetic voltages were generated, we calculated $V_{0,F}$ from $V_{0,M}$ after applying the LR method (see equation 5 of the preprint article). We performed 1000 evaluations of Equation 5 for each set of random values (every set has 10 values, the minimum number of data used for a LR calibration), denoted by $< \tau_{\lambda_M,a} >, \sigma(\tau_{\lambda_M,a}), < \alpha >$ and $\sigma(\alpha)$. Subsequently, we calculated the standard deviation of $V_{0,F}$ obtained from the 1000 evaluations.

The range of values we considered included five values for $< \tau_{\lambda_M,a} >$ (0.02, 0.05, 0.1, 0.25 and 0.5), four values for $< \alpha >$ (0.1, 0.5, 1.0 and 2.0), 100 values for $\sigma(\tau_{\lambda_M,a})$ (ranging from 1 to 20% relative to the average) and 100 values for $\sigma(\alpha)$ (ranging from 1 to 50% relative to the average). These values are consistent with the actual measurements obtained in Valladolid and IZO stations. The analysis has been focused on the CWL pair at 675/500 nm. The results are presented in Figure 1.

In Figure 1, the variability of $V_{0,F}$ is represented on a color map, showing the standard deviation of $V_{0,F}$ relative to the Average ($\sigma(V_{0,F})/ \langle V_{0,F} \rangle$), plotted against the standard deviations of $\tau_{\lambda_M,a}$ and $\alpha$ relative to their averages ($\sigma(\tau_{\lambda_M,a}) /< \tau_{\lambda_M,a} >$ and $\sigma(\alpha)/\langle\alpha\rangle$) for various average values of $\tau_{\lambda_M,a}$ and $\alpha$ ($< \tau_{\lambda_M,a} >$ and $\langle\alpha\rangle$), resulting in a total of 20 subfigures. Panels from left to right correspond to increasing $\langle\alpha\rangle$ values, and panels from up to down correspond with increasing $< \tau_{\lambda_M,a} >$. The variability in $V_{0,F}$ ($\sigma(V_{0,F})/ \langle V_{0,F} \rangle$) is displayed on a logarithmic color scale, where bluer shades indicate lower variability and redder shades indicate higher variability.

In the first place, as expected, the results depicted in the figure show that an increase in any of the different parameters ($< \tau_{\lambda_M,a} >, \sigma(\tau_{\lambda_M,a}), < \alpha >$ and $\sigma(\alpha)$) leads to an increase in the variability of $V_{0,F}$. For clean conditions ($< \tau_{\lambda_M,a} > <=0.02$), the variability of $V_{0,F}$ remains below 1% (except for $< \alpha >= 2$ and $\sigma(\alpha)/\langle\alpha\rangle$ higher that 30%). For very low values of $\langle\alpha\rangle$ ($<=0.1$) and $< \tau_{\lambda_M,a} >$ ($<=0.1$), $\sigma(V_{0,F})/ \langle V_{0,F} \rangle$ remains below 1%, regardless of the variability in $\tau_{\lambda_M,a}$ and $\alpha$ (within the study range). For high values of $\langle\alpha\rangle$ ($>= 1$) and $< \tau_{\lambda_M,a} >$ ($>= 0.25$), $\sigma(V_{0,F})/ \langle V_{0,F} \rangle$ is almost always greater than 1% (except in unrealistic cases where the variability in $\tau_{\lambda_M,a}$ and $\alpha$ is extremely low). For the rest of the intermediate cases, $\sigma(V_{0,F})/ \langle V_{0,F} \rangle$ would generally have values below 10%, reaching lower $\sigma(V_{0,F})/ \langle V_{0,F} \rangle$ values (below 5%) depending on the variability in $\tau_{\lambda_M,a}$

and α. In general, it can be stated that the method should not be applied when $<\tau_{\lambda_M,a}>$ >= 0.25 and $\langle\alpha\rangle$ >= 1.

This information will be included in the supplementary material in the manuscript.

Taking into account this information, we will include the following paragraph in Line 416-419:

"In conclusion, this hybrid calibration technique between the Langley plot reference method and the faster and less accurate Ratio cross-calibration method appears to be a suitable technique for transferring the calibration between instruments with different spectral bands. However, despite being less sensitive to aerosol variations compared to the standard Langley calibration method, the validity of LR relies on the assumption of moderate to low aerosol loads and a moderate to low Ångström exponent during the calibration period, making it unsuitable for cases where $\tau_{a,500} \geq 0.25$ and $\alpha \geq 1.0$."

[Figure]

Figure 1: Colormaps representing $V_{0,F}$ variability as $\sigma(V_{0,F})/\langle V_{0,F}\rangle$ as a function of the standard deviations of $\tau_a$ and $\alpha$ relative to their averages ($\sigma(\tau_{\lambda_M,a})/<\tau_{\lambda_M,a}>$ and $\sigma(\alpha)/\langle\alpha\rangle$) for a set of average values of $\tau_{\lambda_M,a}$ and $\alpha$ ($\langle\alpha\rangle$ = 0.1, 0.5, 1.0, and 2.0) and ($<\tau_{\lambda_M,a}>$ = 0.02, 0.05, 0.1, 0.25, and 0.5) for the 675/500 CWL pair. Panels from left to right correspond to increasing $\langle\alpha\rangle$ values, and panels from top to bottom correspond to increasing $<\tau_{\lambda_M,a}>$ values. $\sigma(V_{0,F})/\langle V_{0,F}\rangle$ is displayed on a logarithmic color scale, where bluer shades indicate lower variability, and redder shades indicate higher variability.

---

## Author Comment (AC2)

Manuscript: Preprint amt-2023-108

**Title: The Langley Ratio method, a new approach for transferring photometer calibration from direct sun measurements**

**Response to Referee#1**

The authors appreciate the overall positive response of the Referee #1 and we would like to thank for his/her constructive comments. In the following, the Referee suggestions (in bold) are in detail addressed (the author's responses are below in blue color).

**The study performed by A. Almansa et al. presents an extension of the current operative method for calibrating the field Cimel instruments from AERONET by calibration transfer, allowing the existence of differences between channels wavelengths from primary and secondary instruments. This is not only useful for improving the transfer when small differences exist between the two channels to be transferred, but also to transfer the calibration from PFR-GAW radiometers, contributing to the traceability of measurements. Also, it allows to apply the cross calibration method to same Cimel models with different nominal channels, such as those from the AERONET-OC type. The results have been validated with standard Langley plots showing good results, and the issues raised have been also addressed.**

**English usage is also clear to my understanding, so I would not propose further need for native English revision.**

**My general recommendation for this manuscript is to be accepted, with minor changes.**

**General comments:**

**- The study would be an extension of a previous work from Fargion (2001), at least for the OC case, but I think the method has also some ideas in common from an older paper from Soufflet (1992). I think it merits to check for it, if the authors didn't do before.**

Soufflet et al. (1992) presented a modified Langley method based on an iterative scheme suitable to perform the calibration transference under changeable aerosol conditions (in terms of type and load), which might occur during the day. Despite being proved to provide excellent results when applied to high AOD conditions in the case of dust storms, improving considerably the performance of the classical Langley calibration method, the authors of this paper think that the goal and physics behind this interesting calibration method are far from the basis of the LR method, presented in this manuscript. Moreover, this method has never been applied to the Cimel or the PFR photometers. In contrast, as we stated in the manuscript, LR method has been proved to be a suitable technique for transferring the calibration between instruments with different spectral bands (Cimels and PFRs) in addition to be a useful tool for detecting and correcting possible instrumental issues in our photometers. However, this reference will be added to the manuscript.

**Specific comments:**

**- Why air mass is limited to minimum 2? Interval 2-5 is common for standard Langleys, but why limiting to air mass 2 in case of cross calibration? For this case, data around noon should be good, even if the airmass is smaller than 2 and some turbulence could make the measurements have higher variability, if this is the reason. Anyway, a comment could be included.**

Yes, as the Referee states, the reason for this limitation is to minimize aerosol variability during the calibration process. If the spectral bands of the two instruments involved were identical, aerosol variability would have a limited impact on the cross-calibration, making data around noon usable without the need for the LR method. However, as the differences in spectral bands increase, aerosol variability has a more significant impact on the calibration results. For locations at low latitudes (latitude < 30º approximately), like Izaña, the optical airmass changes rapidly with time, especially during the summer. Thus, restricting the airmasses to the 2-5 range helps minimize aerosol variability. Including lower airmasses would extend the calibration time, thereby increasing the risk of encountering greater aerosol variability. Conversely, for higher latitude locations in the winter, such as Valladolid, optical airmasses lower than 2 are not even reached, leading to longer calibration times and an increased likelihood of encountering more aerosol variability.

We will add the following sentences in page 8 line 217: "In the same manner as the standard Langley method, we have restricted the optical air mass range from 2 to 5. This reduces the calibration time, especially in low-latitude areas, thereby minimizing the possibility of increased aerosol variability."

**- Page 2, line 27: I think Campanelli et al (2012) would be more meaningful than Campanelli et al. (2004) reference here.**

We agree with this comment. We will introduce this reference in the manuscript.

**- Page 2, line 49: why some associated stations of PFR instruments are not part of GAW? What is the difference with full PFR-GAW stations?**

GAW-PFR stations fulfil the GAW standards and are accepted by the WMO aerosol scientific advisory board. They are owned/calibrated/maintained by PMOD-WRC. Associated stations use PFR instruments belonging to host institutes, they are calibrated at PMOD-WRC and they are not contributing to GAW (it is up to the instrument owners to do so, by applying to GAW). Major requirements for participating in GAW are mentioned here: https://community.wmo.int/en/activity-areas/gaw/research-infrastructure/gaw-stations/procedure-station-inclusion-gaw-programme. There is an exception with the OHP (France) and Valladolid (Spain) stations. In both cases, the PFRs are owned/calibrated/maintained by PMOD-WRC and the main goal is the continuous comparison of PFRs with ACTRIS reference CIMEL instruments, as described in the manuscript.

**- Page 3, line 81: it would be interesting to state main factors causing the higher uncertainty in the ratio cross-calibration.**

We agree with referee comment. The main reason for the higher uncertainty in this method is that, in addition to the uncertainty introduced by the master calibration, we have also the uncertainty during the calibration transfer. The possible factors that can impact on the calibration transfer are: the differences in both spectral bands, the uncertainty in the synchronization of the measurements and the rest of instrumental uncertainties introduced from both instruments (dark current, instrumental noise, tracking, etc.).

We will add the following statement: "…, as the uncertainty depends on the uncertainty of the calibration transfer plus the calibration uncertainty of the master instrument."

**- Page 3, line 86: please add a brief meaning of the CE318-TV12-OC model as it has been introduced here for the first time in the paper.**

The authors agree with the comment, we will add the following piece of text to complete the sentence: "…a modified version of the standard model CE318-T for satellite ocean colour (OC) validation…"

**- Page 4 line 97: not sure the expression "the detectors are filtered" is correct in this case.**

The authors agree with the comment, we will modify the sentence as follows: "The radiation reaching the detectors is filtered…"

**- Page 4, line 103: do the collimator minimize stray light only when the sky radiance is measured?**

We agree with the referee comment, the collimator is always reducing the stray light, but its influence is more important on the sky radiance as it is very low compared to the direct sun radiation. So, we will modify the end of the sentence like this: "…,which is specially necessary for sky radiance measurements near the solar aureole."

**- Page 8, line 214: In the standard Langley method, constant tau is assumed; variations of tau are caused by variations of aerosol burden mainly. In the LR, constant delta_tau is now assumed; what is the main factor for variations of delta_tau during the LR process? I assume AE is a main factor, but a short comment could be useful here.**

We will answer this question by conducting a sensitivity analysis of $\Delta\tau_a$ with respect to AOD and AE.

The aerosol optical depth difference, $\Delta\tau_a$, can be written in terms of the master aerosol optical depth and the Ångström exponent, that is:

$$\Delta\tau_a = \tau_{\lambda_M,a} - \tau_{\lambda_F,a} = \tau_{\lambda_M,a} - \tau_{\lambda_M,a}\left(\frac{\lambda_F}{\lambda_M}\right)^{-\alpha}, \tag{1}$$

where $\tau_{\lambda_M,a}$ is the aerosol optical depth from master instrument, $\tau_{\lambda_F,a}$ is the aerosol optical depth from field instrument and $\alpha$ the Ångström exponent.

We investigated the variability of $\Delta\tau_a$ during the calibration process resulting from variations in $\tau_{\lambda_M,a}$ and $\alpha$. Specifically, we expressed this variability in terms of standard deviation. To accomplish this, we considered various data sets with randomly normal distributed values for $\tau_{\lambda_M,a}$ and $\alpha$. These values were characterized by specific averages and standard deviations. We performed 1000 evaluations of Equation 1 for each set of random values (every set has 10 values, the minimum number of data used for a LR calibration), denoted by $<\tau_{\lambda_M,a}>, \sigma(\tau_{\lambda_M,a}), <\alpha>$ and $\sigma(\alpha)$. Subsequently, we calculated the average and standard deviation of $\Delta\tau_a$.

The range of values we considered included four values for $<\tau_{\lambda_M,a}>$ (0.05, 0.1, 0.25 and 0.5), four values for $<\alpha>$ (0.1, 0.5, 1.0 and 2.0), 100 values for $\sigma(\tau_{\lambda_M,a})$ (ranging from 1 to 20% relative to the average) and 100 values for $\sigma(\alpha)$ (ranging from 1 to 50% relative to the average). These values are consistent with the actual measurements obtained in Valladolid and IZO stations. The analysis has been focused on the CWL pair at 675/500 nm. The results are presented in Figure 1.

[Figure]

Figure 1: Colormaps representing $\Delta\tau_a$ variability as $\sigma(\Delta\tau_a)$ as a function of the standard deviations of $\tau_a$ and $\alpha$ relative to their averages $(\sigma(\tau_{\lambda_M,a})/<\tau_{\lambda_M,a}>$ and $\sigma(\alpha)/\langle\alpha\rangle)$ for a set of average values of $\tau_{\lambda_M,a}$ and

α ($\langle\alpha\rangle$ = 0.1, 0.5, 1.0, and 2.0) and ($< \tau_{\lambda_M,a} >$ = 0.05, 0.1, 0.25, and 0.5) for the 675/500 CWL pair. Panels from left to right correspond to increasing $\langle\alpha\rangle$ values, and panels from top to bottom correspond to increasing ($< \tau_{\lambda_M,a} >$ values. $\sigma(\Delta\tau_a)$ is displayed on a logarithmic color scale, where bluer shades indicate lower variability, and redder shades indicate higher variability.

In Figure 1, the variability of $\Delta\tau_a$ is represented on a color map, showing the standard deviation of $\Delta\tau_a$ ($\sigma(\Delta\tau_a)$), plotted against the standard deviations of $\tau_{\lambda_M,a}$ and α relative to their averages ($\sigma(\tau_{\lambda_M,a})$ /$< \tau_{\lambda_M,a} >$ and σ(α)/$\langle\alpha\rangle$) for various average values of $\tau_{\lambda_M,a}$ and α ($< \tau_{\lambda_M,a} >$ and $\langle\alpha\rangle$), resulting in a total of 16 subfigures. Panels from left to right correspond to increasing $\langle\alpha\rangle$ values, and panels from up to down correspond with increasing $< \tau_{\lambda_M,a} >$. The variability in $\Delta\tau_a$ ($\sigma(\Delta\tau_a)$) is displayed on a logarithmic color scale, where bluer shades indicate lower variability and redder shades indicate higher variability. As a reference, we will assess the results with respect to a threshold $\sigma(\Delta\tau_a)$ value of 0.001 (green color), which corresponds approximately to the middle of the color scale in Figures 1 and 2 of the preprint paper.

In the figure 1, as expected, we can see that an increase in any of the different parameters ($< \tau_{\lambda_M,a} >, \sigma(\tau_{\lambda_M,a}), < \alpha >$ and $\sigma(\alpha)$) leads to an increase in the variability of $\Delta\tau_a$. In general, we can say that for very low values of $\langle\alpha\rangle$ (<=0.1) and $< \tau_{\lambda_M,a} >$ (<=0.1), $\sigma(\Delta\tau_a)$ remains below 0.001, regardless of the variability in $\tau_{\lambda_M,a}$ and α (within the study range). For high values of $\langle\alpha\rangle$ (>=1) and $< \tau_{\lambda_M,a} >$ (>0.1), $\sigma(\Delta\tau_a)$ is almost always greater than 0.001 (except in unrealistic cases where the variability in $\tau_{\lambda_M,a}$ and α is extremely low). For the rest of the intermediate cases, $\sigma(\Delta\tau_a)$ would have values below or above 0.001 depending on the variability in $\tau_{\lambda_M,a}$ and α.

In view of the numerical results, we can say that $\tau_{\lambda_M,a}$ and α have a similar influence on the variability of $\Delta\tau_a$. However, it is true that generally, according to the measurements, α tends to be more variable than $\tau_{\lambda_M,a}$. Therefore, we can conclude that α is the factor that most influences the variability of $\Delta\tau_a$.

This analysis will be extended to the variability of $V_0$ obtained using the LR method to address referee 3 question #7, and it will be included as a supplementary material in the final manuscript.

**- Page 8, equation 6: has been ancillary data used in this equation? or AERONET derived terms for the master instrument? (both tested sites).**

To calculate the optical depth due to Rayleigh scattering and gas absorption (mainly $O_3$ and $NO_2$) we employ the formulas provided in Giles et al. (2019). The ancillary quantities for pressure and gases concentrations are provided in the AERONET data files.

**- Page 8, line 229-230: I think it would be clearer if in this sentence it is stated that tau_F,a is estimated using the Angstrom law using data from the master AOD spectrum (or I assume it is how it has been done).**

The referee is right, we have used the spectral AOD data from the master to estimate $\tau_{f,a}$. Then we have added the following text: "…from the master AOD spectrum…"

**- Page 9, line 267-268: Then no postcalibration and interpolation has been used to get the V_0,SL for the two photometers?**

We have not included post-calibration nor any interpolation of the calibration constants of the master's instruments. This aligns with the typical procedure within the AERONET network for producing final, quality-assured Level 2.0 products. However, as pointed out by Toledano et al. (2018), Cimel masters, such as the instruments employed in this study, exhibit an extraordinarily high level of temporal stability, with an expected degradation of only -0.07% per year. Based on these findings, we can consider the degradation of our master instrument's calibration over a period of six months to be negligible. Consequently, our dataset maintains a quality level suitable for the objectives of this paper, which is to demonstrate the effectiveness of the method in obtaining calibration constants within the operational AERONET uncertainty.

Reference: Toledano, C., González, R., Fuertes, D., Cuevas, E., Eck, T. F., Kazadzis, S., Kouremeti, N., Gröbner, J., Goloub, P., Blarel, L., Román, R., Barreto, Á., Berjón, A., Holben, B. N., and Cachorro, V. E.: Assessment of Sun photometer Langley calibration at the high-elevation sites Mauna Loa and Izaña, Atmos. Chem. Phys., 18, 14555-14567, https://doi.org/10.5194/acp-18-14555-2018, 2018.

**- Page 14, line 340: "tracking problems" make me think of technical problems appearing during tracking. I thknk the authors refer to the general lack of continuous tracking during the measurement sweep, ending on UV wavelengths. Maybe the authors should reformulate somehow the expression, for example "tracking limitations"?**

We agree with the referee comment, we will change the text to "…tracking limitations".

**- Page 15, line 357: what is the measurement time required for a triplet/individual sweep when the pointing is adjusted before each single specxtral measurement?**

The pointing adjustment takes just a fraction of a second, so in total this adjustment adds around 1 or 2 seconds to the standard measurement sequence sequence (about 15 seconds in total for all 10 channels in the filter wheel).

**References suggested:**

**V. Soufflet, C. Devaux, and D. Tanré. Modified langley plot method for measuring the spectral aerosol optical thickness and its daily variations. Appl. Opt., 31(12):2154–2162, 1992.**

**M. Campanelli et al. (2012) Monitoring of Eyjafjallajökull volcanic aerosol by the new European Skynet Radiometers (ESR) network, Atmospheric Environment 48, doi:10.1016/j.atmosenv.2011.09.070.**

---

## Author Comment (AC3)

**Manuscript: Preprint amt-2023-108**

**Title: The Langley Ratio method, a new approach for transferring photometer calibration from direct sun measurements**

**Response to Referee#2**

The authors appreciate the generally positive response from Referee #2, and we would like to thank them for her/his constructive comments. Below, the Referee's suggestions (in bold) are addressed in detail (the authors' responses are provided below in blue).

**This paper entitled present the Langley Ratio method for optimizing the calibration constants between two sun photometer that do not have the same spectral bands, although differences must be minimum. The method is great in advancing the optimization of sun-photometry, particularly between two different networks such as AERONET and GAW-PFR. Authors present the potentiality of the method and its applicability for detecting instrumental drifts. I recommend its publication, but before I have some issues that I would like the authors answer:**

- **Authors claim the importance of different field-of-view (FOV) of the instruments. Can you quantify of this affect the Langley Ratio method.**

  To answer this question, we performed a simulation of irradiance using the radiative transfer code libRadtran (Mayer and Kylling, 2005; Emde et al., 2016), at different air masses (2, 2.5, 3, 3.5, 4, 4.5, and 5), different values of AOD at 500 nm (from 0.02 to 1), and two aerosol types, urban (fine particles predominance) and desert dust (coarse particles), for the different spectral bands and Field of Views (FOVs) of each instrument. To simulate the irradiance for each instrument, we followed the method for simulating circumsolar radiation as described in García et al. 2020. Once the irradiances were simulated, we applied the LR method to obtain the extraterrestrial irradiance of CE318, $E_{0,LR}$, using the PFR as the reference, and compared it with the extraterrestrial irradiance of CE318 using the standard Langley method, $E_{0,SL}$, obtained under low aerosol load conditions.

  The LR method was applied between the nearest pairs of spectral bands, namely 340/368, 380/368, 440/412, 500/500, 675/500, 870/862, 1020/862, and 1640/862. The results obtained are shown in Figure 1. In this figure, the relative difference of extraterrestrial irradiance, $(E_{0,LR} - E_{0,SL})/ E_{0,SL}$, is displayed on a logarithmic scale against the value of AOD at 500 nm, for the two types of aerosols and for each spectral band of CE318. We can see that $E_{0,LR}$ is always higher than $E_{0,SL}$ (except for urban aerosols and $AOD_{500}>0.5$ at 340 nm). We can also observe that urban aerosols have a lower impact than desert aerosols (except for bands 380 and 440 and $AOD_{500}>0.4$), and the difference increases with $AOD_{500}$, being greater for shorter wavelengths. In general, we can say that urban aerosols have a very low impact, whereas desert aerosols have a noticeable impact for moderate to high aerosol loads (from 0.1% to 1% for $AOD_{500}$ between 0.1 and 1, depending on the spectral band).

However, it is important to admit that AOD uncertainty in large particles and high AODs due to FOV and forwards scattered light is much more important than the effect on the method.

[Figure]

**Figure 1:** Relative difference between the extraterrestrial irradiance obtained with the LR method ($E_{0,LR}$) and the SL method ($E_{0,SL}$) against the AOD at 500 nm for two different types of aerosols: urban in blue and desert in orange, for each spectral band of the CE318. The irradiances were simulated using the LibRadtran radiative transfer code, taking into account the Field of Views (FOVs) of the CE318 and the

PFR. The LR method was applied between the closest Central Wavelength (CWL) pairs between the PFR and CE318, namely 340/368, 380/368, 440/412, 500/500, 675/500, 870/862, 1020/862, and 1640/862.

- **I miss an intercomparision between the Langley Ratio and the classical Langley methos. It could have been possible with instruments at Izaña.**

The authors did not consider this possibility because it is only possible to do it at IZO, and we wanted to show the results consistently for both stations. Furthermore, according to Toledano et al 2018, the standard Langley calibration for a single day in IZO has an uncertainty of 0.9%, so it is advisable to average at least 10 calibrations to achieve a lower uncertainty of 0.25%. However, following the referee's suggestion, we carried out the daily comparison of the LR method with the daily SL calibrations for the same time period presented in the article (from July 1, 2021, to December 31, 2022) for each channel of the CE318. The results of this comparison are shown in Figure 2 and Table 1. In this figure, the daily relative difference throughout the study period between calibrations is represented as $(V_{0,LR} - V_{0,SL})/V_{0,SL}$ for each channel of the CE318, differentiating between morning (blue points) and afternoon (orange points). The grey crosses represent the relative differences where SL calibrations do not meet the criteria to consider the calibration as valid (see Toledano et al. 2018). We decided to include these results in the graph to show that on many days, it is not possible to perform a Langley calibration in IZO, particularly in summer due to Saharan dust intrusions. These data are not included in the calculation of the average and standard deviation of the differences presented in Table 1.

In general, we can say that the difference between both calibrations is low, especially for longer wavelengths, with average differences ranging from 0.08% to 0.68% and standard deviations ranging from 0.21% to 1.33%. These results are slightly different from those presented in the article (Figure 1 and Table 1). However, it is important to consider that the data population is different in each case, with 72 data points in the present study compared to 338 in the manuscript.

[Figure]

**Figure 2:** Daily relative differences between the extraterrestrial voltage obtained with the LR method ($V_{0,LR}$) and the SL method ($V_{0,SL}$) at IZO for a six-month period (from 07/01/2021 to 12/31/2021) for each spectral band of the CE318. Blue points represent morning data, orange points represent afternoon data, and grey crosses indicate data that does not meet the SL criteria described in Toledano et al 2018. The LR method was applied to the closest Central Wavelength (CWL) pairs between the PFR and CE318, specifically 340/368, 380/368, 440/412, 500/500, 675/500, 870/862, 1020/862, and 1640/862.

| CE318 spectral band | 1640 | 1020 | 870 | 675 | 500 | 440 | 1020i | 380 | 340 |
|---|---|---|---|---|---|---|---|---|---|
| $\overline{\Delta V_0}$(%) | 0.08 | 0.11 | 0.11 | 0.38 | 0.39 | 0.40 | 0.11 | 0.68 | 0.62 |
| $\sigma(\Delta V_0)$(%) | 0.21 | 0.28 | 0.32 | 0.40 | 0.65 | 0.66 | 0.28 | 1.04 | 1.33 |

**Table 1:** Mean and standard deviation of the daily relative differences (in %) between the standard Langley calibration ($V_{0,SL}$) and the daily Langley ratio calibration ($V_{0,LR}$) at IZO for a six-month period (from 07/01/2021 to 12/31/2021) for the different CE318-TS spectral bands. The LR method was applied to the closest Central Wavelength (CWL) pairs between the PFR and CE318, specifically 340/368, 380/368, 440/412, 500/500, 675/500, 870/862, 1020/862, and 1640/862.

**Technical comments**

- **Abstract: In lines 1-2 you refer to 'photometer'. I propose to specify 'sun photometer'.**

We agree with the referee comment, we will change to sun photometer.

- **Line 44. Can you give the link to GAW-PFR network? Are the data publicly available?**

Yes, the data is publicly available. You can download from the following URL:

https://gawpfr.pmodwrc.ch/ or https://ebas-data.nilu.no/Default.aspx

- **Line 56. What is ACTRIS? Can you give the link?**

Below we reproduce the definition provided in the ACTRIS web page (https://www.actris.eu/):

*"The Aerosol, Clouds and Trace Gases Research Infrastructure (ACTRIS) is the pan-European research infrastructure (RI) producing high-quality data and information on short-lived atmospheric constituents and on the processes leading to the variability of these constituents in natural and controlled atmospheres."*

- **Line 65. 'Langley calibration technique', a reference is needed.**

We agree with the referee comment. We will add the reference from Shaw, 1983.

Shaw, G. E.: Sun photometry, Bull. Am. Meteorol. Soc., 64, 4–10,1983.

- **Line 66. What is the acronym 'SI?**

We missed the acronym definition. SI is for International System of Units. We will add it in the text.

- **Instrumentation: Adding a table that summarizes the main characteristics of each instrument would be ideal. Actually, the authors use different CIMEL versions that have different bands. The reader might find easier the importance of the Langley Ratio technique.**

We will add two tables, one for section 2.1 and another one for section 2.2 describing the CE318-T and the PFR photometers:

Table 1: Main features of the CE318-TS and CE318-TV12-OC sun photometers used in this study.

|  | CE318-TS | CE318-TV12-OC |
|---|---|---|
| Type of instrument | Standard version, Reference instrument in AERONET | Reference instrument in AERONET-OC (ocean color) |
| Type of observation | Automatic sun–sky tracking | Automatic sun–sky–sea tracking |

| Available standard channels | 340, 380, 440, 500, 675 nm, 870, 1020, 1640 nm | 400, 412.5, 442.5, 490, 510, 560, 620, 665, 779, 865, 937, and 1020 nm |
|---|---|---|
| FWHM | 2 nm (340 nm), 4 nm (380 nm), 10 nm (VIS-NIR), 25 nm (1640 nm) | 10 nm |

Table 2: Main features of the CE318-TS and the GAW-PFR sun photometers used in this study

|  | CE318-TS | PFR |
|---|---|---|
| Type of instrument | Standard version, Reference instrument in AERONET | Standard version, Reference instrument in GAW-PFR |
| Type of observation | Automatic sun–sky tracking | Automatic continuous direct sun irradiance |
| Available standard channels | 340, 380, 440, 500, 675 nm, 870, 1020, 1640 nm | 368, 412, 500, 862 nm |
| FWHM | 2 nm (340 nm), 4 nm (380 nm), 10 nm (VIS-NIR), 25 nm (1640 nm) | 5 nm |
| FOV | 1.3° | 2.5° |
| Sun tracker | Robot specifically designed by CIMEL and controlled in conjunction with the radiometer | Any sun tracker with a resolution of at least 0.08° |

- **Lines 110-112: There are more inversion techniques for obtaining aerosol microphysical properties. For example, check GRASP algorithm.**

  We agree with the referee comment, we will add the following references referring to GRASP:

  Dubovik, O., Lapyonok, T., Litvinov, P., Herman, M., Fuertes, D., Ducos, F., Torres, B., Derimian, Y., Huang, X., Lopatin, A., Chaikovsky, A., Aspetsberger, M., and Federspiel, C.: GRASP: a versatile algorithm for characterizing the atmosphere, in: SPIE, vol. Newsroom, 2014.

  Torres, B., Dubovik, O., Fuertes, D., Schuster, G., Cachorro, V. E., Lapyonok, T., Goloub, P., Blarel, L., Barreto, A., Mallet, M., Toledano, C., and Tanré, D.: Advanced characterisation of aerosol size properties from measurements of spectral optical depth using the GRASP algorithm, Atmos. Meas. Tech., 10, 3743–3781, https://doi.org/10.5194/amt-10-3743-2017, 2017.

- **Lines 113-114: A reference is needed for the statement about AOD uncertainties.**

We will add the reference from Eck et al. 1999.

Eck, T., Holben, b., Reid, J., Dubovik, O., Smirnov, A., Neill, Slutsker, I., and Kinne, S.: Wavelength dependence of the optical depth of biomass burning, urban, and desert dust aerosols, Journal of Geophysical Research: Atmospheres, 104, 31 333–31 349, https://doi.org/10.1029/1999JD900923, 1999.

- **Lines 114-117: Will the technique be used for ocean-color applications?**

In our opinion, as the LR technique has been demonstrated to accurately transfer direct sun calibration from a standard CE318-T to a CE318-TV12-OC, it can be valuable for ocean color applications. This point has already been stated in the manuscript, specifically in Line 84 and Lines 397-400.

- **Lines 186-187: I do not understand why Langley technique requires long observational periods of one/two months. The same statement is in the introduction (Lines 76-77). Theoretically with one day of measurements during very clean and stable conditions at high altitude you have Langley calibration.**

In principle, as the referee has pointed out, theoretically, under very pristine conditions, a Langley calibration should be sufficient for accurate instrument calibration. However, in reality, such extremely pristine conditions are not always attainable, even at Langley calibration sites like Izaña. In this regard, Toledano et al. (2018) conducted an analysis and determined that an individual Langley plot at Izaña has an uncertainty of approximately 0.9%. To reduce calibration uncertainty to below 0.25%, these authors concluded that 10 or more Langley plots should be averaged. Achieving such a number of Langley plots would require at least 10 days, assuming ideal conditions, but it can extend to one or two months depending on atmospheric conditions at Izaña, such as the presence of clouds or dust outbreaks, especially during the summer. Such period of observation is also important from the operational point of view, to ensure the stability of the photometers.

- **Equation 3: Is difficult to follow unless you define each of the variable. Same happens for Equation 4.**

We agree with the referee comment, we will clarify it, by changing the text from line 188:

Due to the scarcity of locations with Langley conditions, the typically high costs associated with shipping equipment to such remote areas, and the long time required to conduct this calibration, alternative methods have been developed (Soufflet et al. 1992; Schmid et al. 1998; Holben et al. 1998; Fargion et al. 2001). Specifically, transferring calibration from a Langley-calibrated reference instrument ($V_{0,\lambda}^M$), referred to as the "master," to uncalibrated instruments ($V_{0,\lambda}^F$), known as "field" instruments, conducted in more accessible facilities offers a

practical solution for calibrating multiple instruments simultaneously. In this regard, AERONET applies the method exposed by Holben et al. 1998 and extended by Fargion et al. 2001, where the calibration of the field instrument, $V_{0,\lambda}^{F}$, is determined by calculating the ratio between equation 1 applied to the field instrument and equation 1 applied to the master instrument for measurements that are both coincident in time and within the same spectral band. Consequently, this ratio can be expressed in terms of quasi-coincident ratios between raw direct sun measurements from the master ($V_{\lambda}^{M}$) and the field instrument ($V_{\lambda}^{F}$) as follows:

$$V_{0,\lambda}^{F} = \frac{V_{\lambda}^{F}}{V_{\lambda}^{M}} \cdot V_{0,\lambda}^{M},$$

- **Equation 5: It is not clear to me how do you compute the differences in aerosol optical depth.**

  The differences in aerosol optical depth are calculated as follows:

  $$\Delta\tau_a = \tau_{\lambda_m,a} - \tau_{\lambda_f,a} \approx \tau_{\lambda_m,a} - \left(\frac{\lambda_f}{\lambda_m}\right)^{-\alpha} \cdot \tau_{\lambda_m,a} \approx \tau_{\lambda_m,a} \cdot \left(1 - \left(\frac{\lambda_f}{\lambda_m}\right)^{-\alpha}\right),$$

  where α is the Ångström exponent. Therefore, $\tau_{\lambda_f,a}$ is estimated from $\tau_{\lambda_m,a}$ and α, which are calculated from the master instrument

- **Section 5.1. It is important to know the ranges of AODs you have during the measurement periods.**

  The AOD ranges at 500 nm are between 0.008 and 0.583 for IZO (with an average of 0.093) and between 0.017 and 0.845 (average of 0.123) at Valladolid.

  This information will be added in the manuscript.

- **Line 263: Why limiting to airmasses 2-5**

  The primary reason for this limitation is to reduce the calibration duration and, consequently, minimize aerosol variability. Data beyond airmass 5 are affected by heightened errors in determining optical airmass due to the Earth's spherical shape, in addition to a lower signal-to-noise ratio. Conversely, data below an airmass of 2 change slowly with solar zenith angle, which means it takes longer for the airmass to change. Consequently, this extended period increases the likelihood of encountering greater aerosol variability.

- **Figure 1: How do you explain the outliers in the Figure? Particularly those above 2%. Why positive values predominate?**

  As depicted in Figure 1 of the manuscript, it is evident that most of the higher discrepancies in $V_0$ result from greater variability in Δτ, especially in $\Delta\tau_a$, as aerosols constitute the most variable component within the observed spectral bands. Nevertheless, several other factors can influence the final outcome,

including potential inadequate cloud filtering, the absence of correction for temperature effects in the UV bands of the CE318, or the field of view (FOV) effect. Regarding the prevalence of positive values, we attribute it to the FOV in combination with the increase in AOD. The FOV causes the $V_0$ value obtained with the LR method to increase as AOD rises, as illustrated in Figure 1 of this document.

- **Section 5.3. I do not understand relative differences if you are using the standard Langley calibration. Who is your reference?**

  We assume the referee is referring to the results presented in Figure 4 of the manuscript. In this figure, we depict the relative difference in $V_0$ between the ratio cross calibration and the standard Langley, as well as between the LR (AM and PM) and the standard Langley. In all cases, the standard Langley calibration serves as the reference.

---

## Author Comment (AC4)

**Response to Editor**

The authors have answered all comments by the referees. Although most replies indicate changes and/or additions to the manuscript associated with the comments, a few do not.

- It is not clear if the answer to referee 1 question on post-calibration and interpolation (Page 9, 267-268), is reflected in the modified manuscript. Has the Toledano et al (2018) reference added to the manuscript? It is also not clear if the authors' detailed answer to his/her question on Page 15, line 357 will be reflected in the modified manuscript. Were the references suggested in the referee's last comment added to the manuscript?

  Toledano et al. (2018) is a reference already included in the manuscript since the beginning. The authors will include the following information in lines 267-268:

  "The two $V^{SL}_{0,\lambda}$ values, one per photometer, were considered valid for the six-month period of measurements used in the present study **considering the extraordinarily high temporal stability of Cimel master found by Toledano et al. (2018).**"

- Referee 2 questions on *importance of difference in field of view* and on *inter-comparisons between the LR method and classical Langley methods* were addressed with extensive discussions and graphics. It is not clear, however, how the provided replies are reflected in the revised manuscript:

  The authors performed this calculation to respond accurately to the general comment made by Referee 2. However, our intention is not to include these results in the final manuscript.

- Same question applies to referee 2 comments on line 44:

  Yes, URL will be included in the final manuscript.

- Referee 2 comments on line 56:

  Yes, the authors will include the link to ACTRIS in the manuscript. Once the link is included, an interested reader can access the definition of ACTRIS as well as the main purpose of this Research Infrastructure.

- Referee 2 comments on line 186-187:

  This is a specific question raised by the referee regarding the one to two months observational period required for Langley calibration. The authors do not intend to include this answer in the manuscript, as we believe it is a highly specialized question related to AERONET operations, which is not the primary focus of this paper. We have, however, provided a reference to

Toledano et al. (2018), where all the details of the Langley analysis at Izaña are explained.

- Referee 2 comments on Equation 5:

  Yes, we will include this response in the final manuscript as follows:
  Line 229: "**Specifically, $\Delta\tau_a$ is calculated as follows:**

  $$\Delta\tau_a = \tau_{\lambda_m,a} - \tau_{\lambda_f,a} \approx \tau_{\lambda_m,a} - \left(\frac{\lambda_f}{\lambda_m}\right)^{-\alpha} \cdot \tau_{\lambda_m,a} \approx \tau_{\lambda_m,a} \cdot \left(1 - \left(\frac{\lambda_f}{\lambda_m}\right)^{-\alpha}\right)$$

  **And** $\tau_{\lambda_F,a}$ is estimated using the Angstrom Law (Angstrom, 1929)."

- Referee 2 comments on line 263:

  The authors do not expect to reflect this answer in the manuscript since this is the common air mass range usually selected for Langley calibration in AERONET.

- Referee 2 comments on section 5.3.:

  The authors will include the following information in the Figure 4 caption:

  "$V_0$ relative differences (in %) between the calibration constant obtained by applying the standard Langley calibration ($V^{SL}_0$) using a CE318-TS at Izaña **as the reference** and the calibration constant transferred between two CE318-TS using the Ratio method ($V^R_0$) and the LR method ($V^{LR}_0$) to daily observations at Izaña for 340 nm".

- Similar observation applies to reply to referee 3 comment on Line 70 of the manuscript:

  The authors do not expect to include this answer in the final manuscript since we consider that this fact is clearly stated in the manuscript. For example in lines 202-203 we can read:

  "These assumptions and the common Ratio cross-calibration method itself are valid as long as the spectral bands for both photometers are very similar (i.e. $\Delta\lambda \sim 0$ and similar FWHM)."

  So, the LR method has been created specifically for those conditions in which $\Delta\lambda$ and FWHM difference are relevant, as stated in the text.